# The *Salmonella* virulence protein MgtC promotes phosphate uptake inside macrophages

Soomin Choi[1,5], Eunna Choi[1,5], Yong-Joon Cho[2,5], Daesil Nam[3], Jangwoo Lee[4] & Eun-Jin Lee [1]

The MgtC virulence protein from the intracellular pathogen *Salmonella enterica* is required for its intramacrophage survival and virulence in mice and this requirement of MgtC is conserved in several intracellular pathogens including *Mycobacterium tuberculosis*. Despite its critical role in survival within macrophages, only a few molecular targets of the MgtC protein have been identified. Here, we report that MgtC targets PhoR histidine kinase and activates phosphate transport independently of the available phosphate concentration. A single amino acid substitution in PhoR prevents its binding to MgtC, thus abrogating MgtC-mediated phosphate transport. Surprisingly, the removal of MgtC's effect on the ability to transport phosphate renders *Salmonella* hypervirulent and decreases a non-replicating population inside macrophages, indicating that MgtC-mediated phosphate transport is required for normal *Salmonella* pathogenesis. This provides an example of a virulence protein directly activating a pathogen's phosphate transport inside host.

[1] Department of Life Sciences, College of Life Sciences and Biotechnology, Korea University, Seoul 02841, South Korea. [2] Korea Polar Research Institute, Incheon 21990, South Korea. [3] Division of Microbiology, Department of Molecular Cell Biology, Samsung Biomedical Research Institute, Sungkyunkwan University School of Medicine, Suwon 16419, South Korea. [4] Department of Genetic Engineering and Graduate School of Biotechnology, College of Life Sciences, Kyung Hee University, Yongin 17104, South Korea. [5] These authors contributed equally: Soomin Choi, Eunna Choi, Yong-Joon Cho. Correspondence and requests for materials should be addressed to E.-J.L. (email: eunjinlee@korea.ac.kr)

Phosphorus is one of the most abundant elements found in living organisms. As in the form of phosphate ions, it is found in lipids, nucleic acids, proteins, or carbohydrates; participates in many enzymatic reactions relying on the transfer of phosphoryl group; serves as a reservoir of chemical energy when incorporated into ATP or linked to other molecules. In bacteria, phosphate signaling is mediated by the PhoB/PhoR two-component regulatory system (the PhoP/PhoR for gram-positive bacteria), which has been most extensively studied in *Escherichia coli*[1,2]. PhoR is a histidine kinase that likely responds to the periplasmic phosphate concentration. PhoR is anchored to the inner membrane by an N-terminal transmembrane-spanning region (TM), which is followed by four cytoplasmic domains: a charged region (CR), a Per-ARNT-Sim (PAS) domain, a dimerization/histidine phosphotransfer (DHp) domain, and a catalytic/ATP-binding (CA) domain[1]. Like other histidine kinases, PhoR phosphorylates a histidine residue at the DHp domain and then transfers the phosphoryl group to an aspartic residue of the cognate PhoB response regulator when the periplasmic phosphate levels are low (<4 $\mu$M)[2]. The phospho-PhoB binds to the promoter regions of several genes, such as *phoE*, *pstSCABphoU*, *ugpBAECQ*, *phoBR*, and *phnSTUV*, encoding an outer membrane phosphate porin, a high-affinity phosphate transporter, a glycerol-3-phosphate ABC transporter, the PhoB/PhoR two-component regulatory system, and an aminoethylphosphonate transporter, respectively, and activates transcription of Pho regulon involved in phosphate transport and homeostasis[2]. However, unlike other histidine kinases, PhoR lacks a periplasmic-binding domain that is required to sense a periplasmic signal(s), raising the possibility that phosphate sensing via PhoR may occur in the cytoplasm. This notion is further supported by the presence of the cytoplasmic PAS domain, which is known to be involved in the interaction with other proteins/small molecules or signal sensing[3,4]. In addition, because the PhoU peripheral protein links the PAS domain of PhoR and PstB in the PstSCAB high-affinity phosphate transporter[5], it has been suggested that phosphate signaling via the PhoB/PhoR two-component system might be influenced by the activity of the PstSCAB transporter. Yet, the nature of the signal(s) or the mechanism by which bacteria sense and transduce a cytoplasmic signal(s) via this atypical histidine kinase is unclear given that the intracellular phosphate concentration is relatively constant[6,7]. In this paper, we identify a specific condition that controls PhoR histidine kinase and show that this identified PhoR regulation is required for *Salmonella* pathogenesis.

The intracellular pathogen *Salmonella enterica* serovar Typhimurium has the ability to survive and replicate within a macrophage phagosome, which leads to cause a lethal disease[8]. This bacterium's ability to survive and replicate inside macrophages is largely dependent on production of many virulence proteins during infection. MgtC is one such protein that is critical for intramacrophage survival and MgtC's function for macrophage survival is highly conserved in unrelated intracellular pathogens, such as *Mycobacterium tuberculosis*, *Yersinia pestis*, *Burkholderia cenocepacia*, and *Brucella suis*[9–13]. Previous studies have reported several MgtC targets[14–17]. Among them, MgtC interacts with the membrane-embedded *a* subunit of the $F_1F_o$ ATP synthase and thus inhibits proton translocation across the bacterial inner membrane and ATP production[14]. In turn, this property promotes *Salmonella*'s pathogenicity by maintaining the bacterium's cytosolic pH against a large number of protons generated during phagosome acidification[14]. Consistent with the proposed function in intramacrophage survival, *mgtC* expression is highly induced inside macrophages[18]. High levels of *mgtC* expression are mediated by transcriptional regulation taking place at two different levels. At transcription initiation, the PhoP/PhoQ two-

component system controls *mgtC* expression and increases messenger RNA (mRNA) levels in response to low $Mg^{2+}$, acidic pH, and antimicrobial peptides[19–22], which might be relevant in a macrophage phagosome. In addition, at transcription elongation, the preceding 296-nt leader RNA further increases *mgtC* expression in response to an increase in ATP levels and a decrease in charged-tRNA$^{Pro}$ levels[23–26].

Although MgtC's binding to the $F_1F_o$ ATP synthase enables *Salmonella* to deal with an acidic environment inside a phagosome, this seems to be a radical idea because MgtC production results in a decrease in ATP levels and membrane potential[14] and also because most of the nutrient uptake and ion transport through the membrane require ATP or membrane potential as an energy source. Therefore, it is reasonable that *Salmonella* limits the amount of the MgtC protein by producing the AmgR antisense RNA to degrade *mgtC* mRNA[27], the MgtR peptide to guide MgtC proteolysis[17], and the CigR protein to sequester the MgtC protein at the early stage of infection[15]. At the same time, *Salmonella* needs to have additional strategies to deliver nutrients or ions into the bacterium inside macrophages. Here, we identify a mechanism of the MgtC virulence protein that activates phosphate transport inside macrophages. MgtC targets PhoR histidine kinase and promotes PhoR's autophosphorylation activity, resulting in an increase in phosphate uptake. We determined that a single amino acid substitution in PhoR preventing MgtC's binding renders *Salmonella* hypervirulent and decreases non-replicating *Salmonella* inside macrophages, suggesting that MgtC's interaction with PhoR histidine kinase is required for the normal course of *Salmonella* infection. Our findings provide an example of a *Salmonella* virulence protein that targets a sensor kinase to activate phosphate transport even without sensing a cognate signal.

## Results

***mgtC* overexpression promotes mRNA levels of Pho genes**. To explore other targets of the MgtC protein, we used a *Salmonella* strain harboring a plasmid with the *mgtC* coding region expressed from an IPTG-inducible promoter. Then, we compared the RNA profiles of *Salmonella* expressing the *mgtC* ORF and those expressing the vector after IPTG induction for 1 h. Surprisingly, we identified that mRNA levels of 72 genes including the *mgtC* gene itself were significantly higher in *Salmonella* expressing the *mgtC* gene than those expressing the vector (≥4-fold; Fig. 1b and Supplementary Table 3). By contrast, mRNA levels of ten genes were downregulated in *mgtC*-expressing *Salmonella* (≥4-fold; Fig. 1b and Supplementary Table 3). Among the upregulated genes, we were particularly interested in 17 genes whose functions are involved in phosphate transport/homeostasis (Pho regulon), because little is known about the molecular connection between the MgtC virulence protein and Pho regulon. These genes include the *phoE* gene, *pstSCABphoU* operon, *ugpBAECQ* operon, *phoBR* operon, and *phnSTUV* operon. Elevated mRNA levels of each gene by expressing the *mgtC* gene were verified by quantitative real-time PCR (Fig. 1c–h). As control experiments, mRNA levels of the *phoP* and *mgtB* genes were unaffected by *mgtC* overexpression (Supplementary Fig. 1).

**MgtC interacts with PhoR histidine kinase**. As *mgtC* overexpression increases mRNA levels of the Pho regulon, we suspected that the *mgtC* gene might have an impact on the regulatory system that controls expression of Pho genes. In *E. coli*, the PhoB/PhoR two-component system controls transcription of genes involved in phosphate transport[1]. If this is true for *Salmonella* and MgtC affects expression of these identified genes via the PhoB/PhoR two-component system, we hypothesized that we

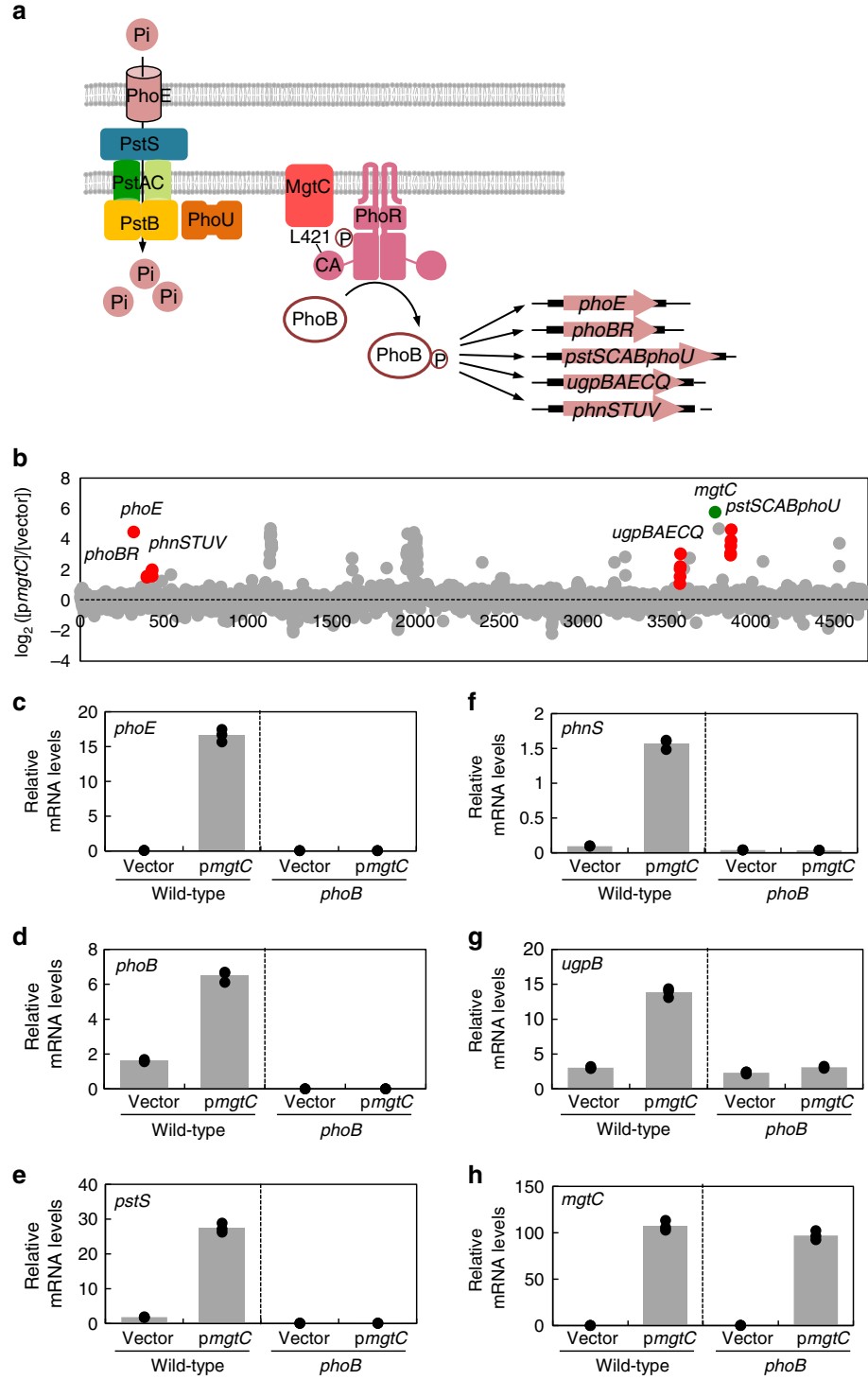

would not observe the increase in mRNA levels of these genes by *mgtC* overexpression in a *phoB* mutant. As expected, *mgtC* overexpression could not promote mRNA levels of the genes identified above in the *phoB* mutant (Fig. 1c–h), suggesting that MgtC increases mRNA levels of Pho regulon by impacting the signaling via the PhoB/PhoR two-component system.

Phosphate signaling might include additional components besides the PhoB/PhoR two-component system because the cytoplasmic PhoU protein links the PhoR histidine kinase and PstB protein in the PstSCAB high-affinity phosphate transporter, thus suggesting a possible engagement of the PhoU protein and PstSCAB transporter in phosphate signaling (Fig. 2a)[5]. To

understand how MgtC promotes transcription of Pho regulon, we investigated whether MgtC physically interacts with one of the above proteins. We used a bacterial two-hybrid assay to assess the interaction between MgtC and phosphate signaling proteins by expressing a C-terminal fusion of the *cyaA*-T18 fragment to the *mgtC* gene and N-terminal fusions of the *cyaA*-T25 fragment to the *phoR*, *phoU*, *pstB*, and *pstA* genes in an *E. coli* strain lacking the CyaA adenylate cyclase. (Here, we focused on the interaction between MgtC and proteins either located in or associated with the inner membrane because MgtC is an inner membrane protein.) We then spotted cells on an LB plate containing X-gal and measured β-galactosidase production from

**Fig. 1** Regulation of the *Salmonella* PhoB/PhoR two-component system by the MgtC virulence protein. **a** When external phosphate levels are low, PhoR histidine kinase phosphorylates the PhoB response regulator. Phosphorylated PhoB activates transcription of several genes involved in phosphate transport including the PhoB/PhoR two-component system itself, PhoE outer membrane phosphate channel, PstSCAB inner membrane phosphate transporter, PhoU-negative regulator, UgpBAECQ glycerol-3-phosphate ABC transporter, and PhnSTUV aminoethylphosphonate transporter. Independently of available phosphate levels, the MgtC virulence protein binds to the Leu421 residue of PhoR histidine kinase, activates autophosphorylation of the PhoR protein, and subsequently induces transcription of the cognate PhoB-dependent genes. This results in an increase in phosphate transport even in the absence of low-phosphate signal. **b** Heterologous expression of the *mgtC* gene promotes mRNA levels of genes involved in phosphate transport. Scatter plot analysis of RNA sequencing in the *mgtC*-expressing versus the vector-expressing *Salmonella*. Genes involved in phosphate transport are indicated in red. The *mgtC* gene from the plasmid is indicated in green. **c**–**h** Relative mRNA levels of the *phoE* (**c**), *phoB* (**d**), *pstS* (**e**), *phnS* (**f**), *ugpB* (**g**), and *mgtC* (**h**) genes in either wild-type (14028s) or a *phoB* deletion mutant *Salmonella* (KK10) harboring a plasmid with the *mgtC* gene (p*mgtC*) or the vector (pUHE21-2lacI$^q$). Bacteria were grown for 3 h in N-minimal media containing 10 mM Mg$^{2+}$ and then for an additional hour in the same media containing 0.5 mM Mg$^{2+}$ and 0.25 mM IPTG. Shown are the means and SD ($n = 3$, independent measurements). Relative mRNA levels represent (target RNA/*rrsH* RNA) ×10,000. See also Supplementary Fig. 1

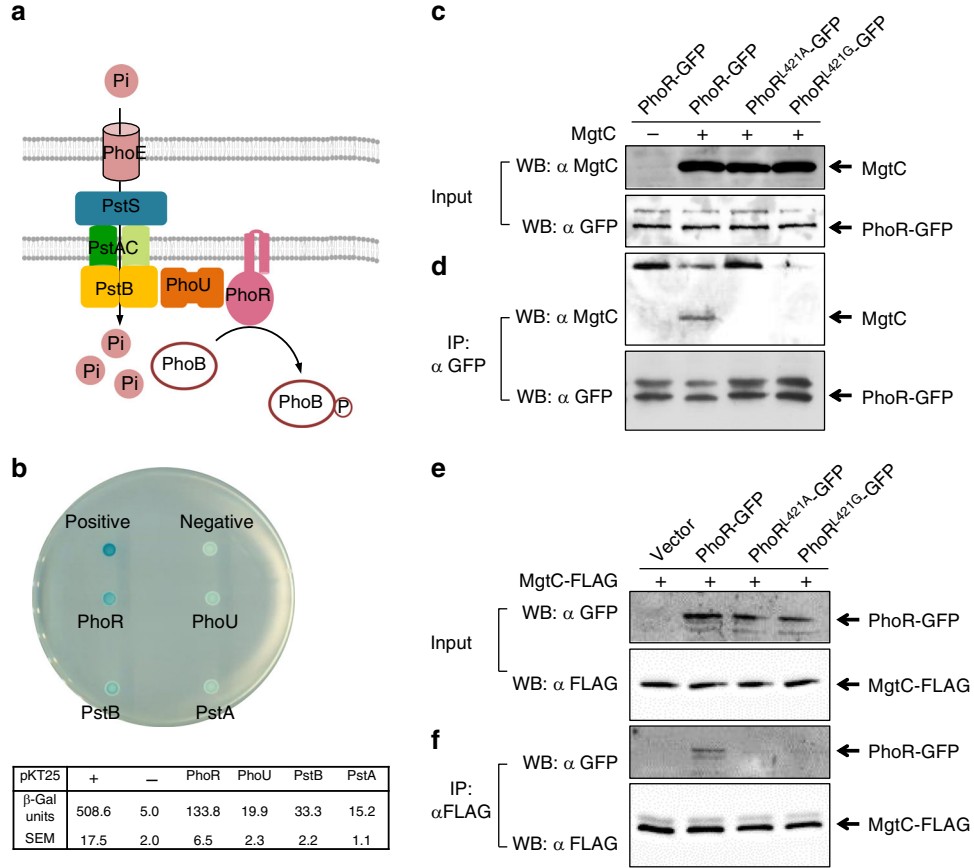

| pKT25 | + | — | PhoR | PhoU | PstB | PstA |
|---|---|---|---|---|---|---|
| β-Gal units | 508.6 | 5.0 | 133.8 | 19.9 | 33.3 | 15.2 |
| SEM | 17.5 | 2.0 | 6.5 | 2.3 | 2.2 | 1.1 |

**Fig. 2** MgtC interacts with the PhoR histidine kinase. **a** Schematic representation of seven components of proteins involved in phosphate signaling and the PhoE outer membrane porin. **b** Bacterial two-hybrid assay between MgtC and proteins involved in phosphate signaling. *Escherichia coli* BTH101 strains harboring two plasmids, pUT18 and pKT25 derivatives expressing a C-terminal fusion of the *cyaA* T18 fragment to the *mgtC* coding region and N-terminal fusions of the *cyaA* T25 fragment to either the coding regions of the *phoR* (PhoR), *phoU* (PhoU), *pstB* (PstB), *pstA* (PstA), and *mgtR* (positive) genes or the pKT25 empty vector (negative) were spotted onto LB plates containing 80 μM X-Gal and 0.1 mM IPTG and incubated at 30 °C for 40 h. Blue colored colonies indicate a positive interaction. The average β-galactosidase activities (β-Gal units) are shown below with the SEM ($n = 3$, independent measurements). **c**, **d** The C-terminally GFP-tagged PhoR protein immunoprecipitates MgtC. **c** Crude extracts prepared from wild-type *Salmonella* coexpressing PhoR-GFP, PhoR$^{Leu421Ala}$-GFP, or PhoR$^{Leu421Gly}$-GFP and either MgtC or the empty vector (pBAD33) were detected with anti-MgtC (upper) and anti-GFP (lower) antibodies. **d** Eluted fractions prepared from strains listed above were detected with anti-MgtC (upper) and anti-GFP (lower) antibodies after immunoprecipitation with anti-GFP antibody-coated beads. **e**, **f** The C-terminally FLAG-tagged MgtC protein immunoprecipitates PhoR-GFP. **e** Crude extracts prepared from wild-type *Salmonella* coexpressing PhoR-GFP, PhoR$^{Leu421Ala}$-GFP, or PhoR$^{Leu421Gly}$-GFP, or the empty vector (pTGFP) and MgtC-FLAG were detected with anti-GFP (upper) and anti-FLAG (lower) antibodies. **f** Eluted fractions prepared from strains listed above were detected with anti-GFP (upper) and anti-FLAG (lower) antibodies after immunoprecipitation with anti-FLAG antibody-coated beads. Bacteria were grown for 3 h in N-minimal media containing 10 mM Mg$^{2+}$ and then for an additional hour in the same media containing 0.5 mM Mg$^{2+}$ and 1 mM L-arabinose and crude extracts were prepared as described in Methods

a cAMP-dependent promoter that could be produced when T18 and T25 fragments of the *cyaA* gene functionally complemented by a physical interaction between fused MgtC and the partner proteins. Among the strains we tested, the strain expressing MgtC-T18 and T25-PhoR showed a strong blue color on the LB X-gal plate, indicating that MgtC interacts with PhoR histidine kinase (Fig. 2b). By contrast, other proteins coexpressed with MgtC-T18 exhibited a weak or no interaction, similar to those coexpressing the empty T25 fragment (Fig. 2b). The MgtR peptide that interacts with MgtC[17,28], was used as a positive control.

As an independent approach, we verified the interaction between MgtC and PhoR using an immunoprecipitation assay. A C-terminally GFP-fused PhoR protein immunoprecipitated MgtC in protein extracts prepared from a strain expressing PhoR-GFP and MgtC from a constitutive and an arabinose-inducible promoter, respectively (Fig. 2c, d). Likewise, C-terminally FLAG-tagged MgtC immunoprecipitated PhoR-GFP (Fig. 2e, f). These experiments demonstrated that MgtC interacts with PhoR histidine kinase.

**Leu421 of the PhoR protein is required for MgtC interaction.** Given that Asn92 in MgtC is required for interaction with both PhoR and $F_1F_o$ ATP synthase[14] (Supplementary Note and Supplementary Fig. 2), we started to search for a region or residue of PhoR that is required for MgtC interaction to assess the effect of MgtC binding to PhoR independently of its binding to $F_1F_o$ ATP synthase. To identify a region(s) of PhoR that interacts with MgtC, we constructed several T25-fused *phoR* fragments corresponding to the TM, PAS including the short CR, CA, or both the TM and PAS domains, and tested whether each domain could interact with coexpressed MgtC-T18 (Supplementary Fig. 3a). None of the domains exhibited β-galactosidase activity when coexpressed with MgtC-T18 (Supplementary Fig. 3b), suggesting that none of the domains in the PhoR protein are sufficient for MgtC interaction. We then created a series of T25-fused *phoR* derivatives deleted from their C-termini to determine the minimal requirement for MgtC interaction. Interestingly, *phoR* derivatives that harbor the coding region up to the amino acid 421 or more (421, 422, 423, 424, 425, 426, or full-length *phoR*) retained the ability to interact with MgtC, while *phoR* derivatives with coding regions up to amino acid 420 or less (420, 419, 418, or 417) lost this ability (Fig. 3a). These results indicate that the amino acid at position 421 in PhoR, which corresponds to leucine (Fig. 3b), is required for MgtC interaction.

Homology-based structure modeling suggested that Leu421 is located at the end of the fourth β-stand in the CA domain, which is followed by a disordered C-terminal region (422–431) (Fig. 3c). T25-PhoR variants in which the Leu421 residue is substituted by residues with smaller side chains, such as valine, alanine, and glycine lost the ability to interact with MgtC-T18, whereas a T25-PhoR variant with the Leu421 to Ile substitution retained a weak interaction (Supplementary Fig. 4), indicating that the size of the hydrophobic side chain of Leu421 is critical for MgtC interaction.

**MgtC promotes autophosphorylation of PhoR histidine kinase.** The CA domain of bacterial histidine kinase is involved in ATP binding and catalyzes the transfer of γ-phosphate from ATP to a key histidine residue in the DHp domain (His213 for *Salmonella* PhoR), a process called autophosphorylation[29]. As the position of Leu421 appears to be in parallel to that of His213 in the DHp domain (Fig. 3d), the location and configuration of the leucine residue led us to hypothesize that MgtC's binding would promote autophosphorylation by making ATP accessible to His213 in the DHp domain. To test this, we prepared membrane fractions from

wild-type cells expressing either PhoR alone or MgtC and PhoR together and measured the accumulation rate of phospho-PhoR after adding γ-radiolabeled ATP. Membrane fractions prepared from wild-type cells expressing both MgtC and PhoR accumulated PhoR-P faster than those prepared from cells expressing PhoR alone (Fig. 3e, f), supporting our hypothesis that MgtC promotes autophosphorylation of PhoR histidine kinase by binding to the Leu421 residue of PhoR.

***phoR*[L421A] substitution prevents MgtC's effect on PhoR.** Given that all experimental conditions tested above contain 10 mM phosphate, MgtC seems to promote PhoR autophosphorylation even when phosphate is not limiting. To get a further insight into the MgtC-mediated control of PhoR histidine kinase, we created a *phoR* chromosomal mutant where Leu421 was substituted by either the Ala or Gly (*phoR*[L421A] to *phoR*[L421G])(Fig. 4a), lacking the ability to interact with MgtC (Fig. 2c–f, and Supplementary Fig. 4). The *phoR*[L421A] to *phoR*[L421G] substitutions were assumed to disrupt MgtC's binding to PhoR independently of its binding to $F_1F_o$ ATP synthase. Therefore, we first checked whether these substitutions affect intracellular ATP levels. Intracellular ATP levels of the *phoR* substitution mutants were unaffected in all tested conditions (Supplementary Fig. 5), suggesting that these *phoR* substitutions have no effect on the interaction between MgtC and $F_1F_o$ ATP synthase.

Next, because the *mgtC* gene is highly expressed in low $Mg^{2+}$ but repressed in high $Mg^{2+}$ by the PhoP/PhoQ two-component system[19], we expected that we could observe MgtC's regulatory action on PhoR only in low $Mg^{2+}$ but not high $Mg^{2+}$. The following results demonstrate our hypothesis: First, the levels of PhoR-P in the membrane vesicles prepared from wild-type cells increased immediately after the addition of γ-labeled ATP in low $Mg^{2+}$ (Fig. 4b), but PhoR-P was not detected until 15 min in high $Mg^{2+}$ (Fig. 4e). This is due to MgtC's binding to PhoR, because the *phoR*[L421A] to *phoR*[L421G] substitutions prevented the increase of PhoR-P levels in low $Mg^{2+}$ (Fig. 4c, d). Control experiments proved that PhoR-P levels were unaffected by the *phoR*[L421A] to *phoR*[L421G] substitutions in high $Mg^{2+}$ (Fig. 4f, g).

In agreement with elevated PhoR-P levels observed in low $Mg^{2+}$, *Salmonella* grown in low $Mg^{2+}$ promoted expression of Pho regulon because wild-type *Salmonella* increased the amounts of the HA-tagged PhoB protein (Fig. 4h) and mRNA levels of the *phoE* gene in low $Mg^{2+}$ (Fig. 4k). The MgtC-mediated increases of the PhoB-HA protein and *phoE* mRNA were not observed in the *phoR*[L421A] to *phoR*[L421G] mutants (Fig. 4h, k). Control experiments were carried out as follows: the increases in the PhoB-HA protein levels and *phoE* mRNA levels were not detected in non-inducing media containing 10 mM $Mg^{2+}$ (Fig. 4h, k), the *phoR*[L421A] to *phoR*[L421G] substitutions did not affect the expression behaviors of mRNA or protein levels of the *mgtC* gene (Fig. 4i, l), and Fur protein levels were unaffected in all tested conditions (Fig. 4j). We pursued the *phoR*[L421A] substitution for further experiments because even though the *phoR*[L421A] substitution eliminates MgtC-mediated PhoR autophosphorylation, the *phoR*[L421A] variant appears to be fully functional in terms of phosphate signaling, given that it retained the ability to produce PhoR-P and promote *phoE* mRNA levels in low phosphate, a PhoB/PhoR-inducing condition (Fig. 4k). It is interesting to note that the *phoR*[L421G] substitution had no effect on PhoR autophosphorylation but showed a defect in the increase of *phoE* mRNA levels in low phosphate (Supplementary Fig. 6 and Fig. 4k), suggesting that it might compromise the ability of PhoR-P to transfer its phosphate to PhoB.

The fact that MgtC increases PhoR-P levels and thus induces mRNA levels of Pho regulon suggests that the *phoR*[L421A]

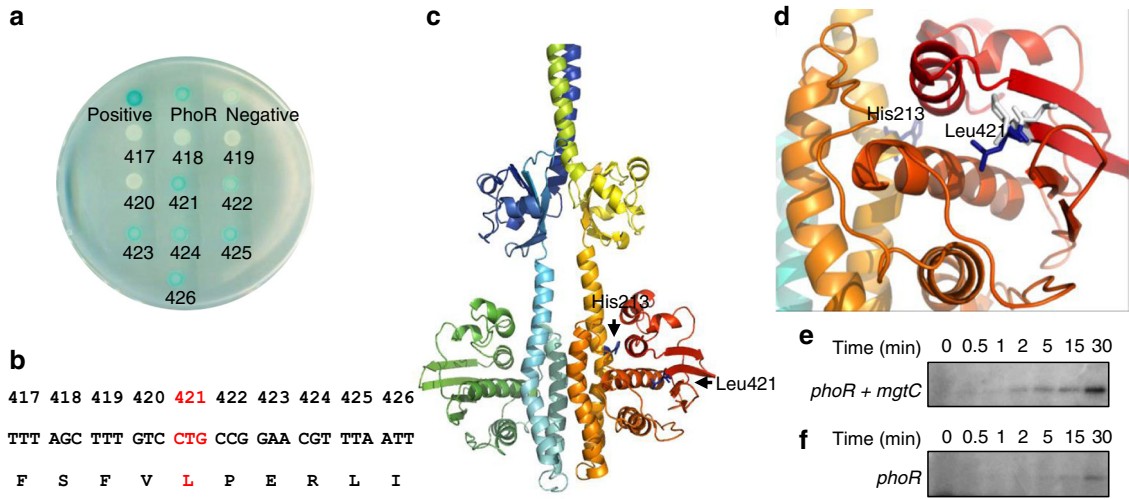

**Fig. 3** Leu421 of the PhoR protein is required for MgtC interaction. **a** Bacterial two-hybrid assay between the MgtC and wild-type PhoR protein or its derivatives. *Escherichia coli* BTH101 strains harboring two plasmids, pUT18 and pKT25 derivatives expressing the C-terminal fusion of the *cyaA* T18 fragment to the *mgtC* coding region and N-terminal fusions of the *cyaA* T25 fragment to either the coding regions of the full-length *phoR* (PhoR), *phoR*₁₋₄₁₇ (417), *phoR*₁₋₄₁₈ (418), *phoR*₁₋₄₁₉ (419), *phoR*₁₋₄₂₀ (420), *phoR*₁₋₄₂₁ (421), *phoR*₁₋₄₂₂ (422), *phoR*₁₋₄₂₃ (423), *phoR*₁₋₄₂₄ (424), *phoR*₁₋₄₂₅ (425), *phoR*₁₋₄₂₆ (426), and *mgtR* (positive) genes or the pKT25 empty vector (negative) were indicated. Cells were spotted onto LB plates containing 80 μM X-Gal and 0.1 mM IPTG and incubated at 30 °C for 40 h. Blue colored colonies indicate a positive interaction. See also Supplementary Fig. 3. **b** The amino acid sequence of the PhoR protein corresponding to amino acids 417–426. **c** A modeled structure of the PhoR dimer predicted from a homology-modeling program (Phyre2) based on the crystal structure of *Streptococcus mutans* VicK (PDB: 4I5S). Leu421 is located at the outer surface of the CA domain and is in parallel to His213 in the DHp domain, which is required for autophosphorylation. The positions of Leu421 in the CA domain and His213 in the DHp domain are indicated by arrows. **d** A magnified side view of PhoR in **c** including Leu421 in the CA domain and His213 in the DHp domain. The Leu421 and His213 residues are indicated with blue sticks and Leu421-neighboring residues (Val420 and Pro422) are indicated with pale gray sticks. See also Supplementary Fig. 4. **e**, **f** Autophosphorylation assay to determine the rate of PhoR phosphorylation. Levels of PhoR-P following incubation of membrane vesicles prepared from *Salmonella* expressing PhoR in the presence (**e**) or absence of MgtC (**f**) with [γ³²P] ATP at the indicated times. Bacteria were grown for 3 h in N-minimal media containing 10 mM Mg²⁺ and then for an additional hour in the same media containing 0.5 mM Mg²⁺, 0.25 mM IPTG, and 1 mM L-arabinose and membrane vesicles were prepared as described in Methods

substitution might have an impact on *Salmonella*'s ability to transport phosphate. *Salmonella* expressing MgtC from the plasmid immediately increased uptake of ³²P-labeled orthophosphate from the medium (Supplementary Fig. 7a). The increase in phosphate uptake is mediated by MgtC's binding because the *phoR*^L421A variant did not increase phosphate uptake in the presence of MgtC (Supplementary Fig. 7b). The vector-expressing *Salmonella* did not show uptake of ³²P-labeled orthophosphate in either the wild-type or the *phoR*^L421A substitution mutant (Supplementary Fig. 7). Similar to what we observed in *Salmonella* strains expressing MgtC from the plasmid, wild-type *Salmonella* started to accumulate inorganic phosphate immediately after the addition of ³²P-labeled orthophosphate when grown in low Mg²⁺ to express MgtC from its chromosomal location (Fig. 4m). However, the *phoR*^L421A variant was defective in phosphate transport in the same media (Fig. 4m). Control experiments demonstrated that both cells did not increase phosphate transport when grown in non-inducing media containing 10 mM Mg²⁺ (Fig. 4n).

**MgtC-PhoR binding is required for Pho expression inside host.** Previous transcriptome analyses have reported that several genes controlled by the PhoB/PhoR two-component system are highly induced when *Salmonella* is inside macrophages[18,30]. We speculated that such an increase in the mRNA levels of the PhoB/PhoR-controlled genes during infection might be due to the presence of the MgtC protein because the *mgtC* gene is one of the most highly expressed genes inside macrophages[18], and also because MgtC activates PhoR autophosphorylation to promote PhoB-dependent gene expression (Figs. 1 and 4). To explore this, we infected *Salmonella* strains into the macrophage-like cell line

J774A.1 and measured mRNA levels of the *phoE* and *phoB* genes (Fig. 1c, d). In wild-type *Salmonella*, mRNA levels of both the *phoE* and *phoB* genes increased at 2 h, peaked at 6 h, slightly decreased at 9 h, and then further decreased at 21 h post infection (Fig. 5a, b). We ascribed these expression behaviors to MgtC's action on PhoR because (i) the mRNA profiles of the *phoE* and *phoB* genes exhibited similar patterns to that of the *mgtC* gene (Fig. 5a–c) and (ii) either a disruption of the MgtC–PhoR interaction by introducing the *phoR*^L421A substitution or a removal of the *mgtC* gene eliminated the increase in the mRNA levels of both the *phoE* and *phoB* genes inside macrophages (Fig. 5a–c). These data suggest that the interaction between MgtC and PhoR is required for a full induction of the mRNA levels of the *phoE* and *phoB* genes inside macrophages.

***phoR*^L421A substitution promotes *Salmonella*'s virulence.** We then speculated whether MgtC-mediated expression of the Pho regulon has a physiological consequence(s) during *Salmonella* infection. To explore this, we measured the replication efficiency of *Salmonella* strains within macrophages. Unexpectedly, the *phoR*^L421A substitution that prevents MgtC interaction increased *Salmonella*'s survival inside macrophages by up to ~300%, compared to that observed in wild-type *Salmonella* at 18 h post infection (Fig. 5d). Such an increase is likely due to the inability of *Salmonella* to express Pho genes because the *phoB* mutant responsible for transcription of the Pho genes showed a similar increase in intramacrophage survival (Fig. 5d). As previously reported[14,31], the *mgtC* mutant showed a severe defect in macrophage survival (Fig. 5d). MgtC's virulence function is mostly due to MgtC's inhibitory action on the F₁F₀ ATP synthase because both the *atpB* mutant lacking the

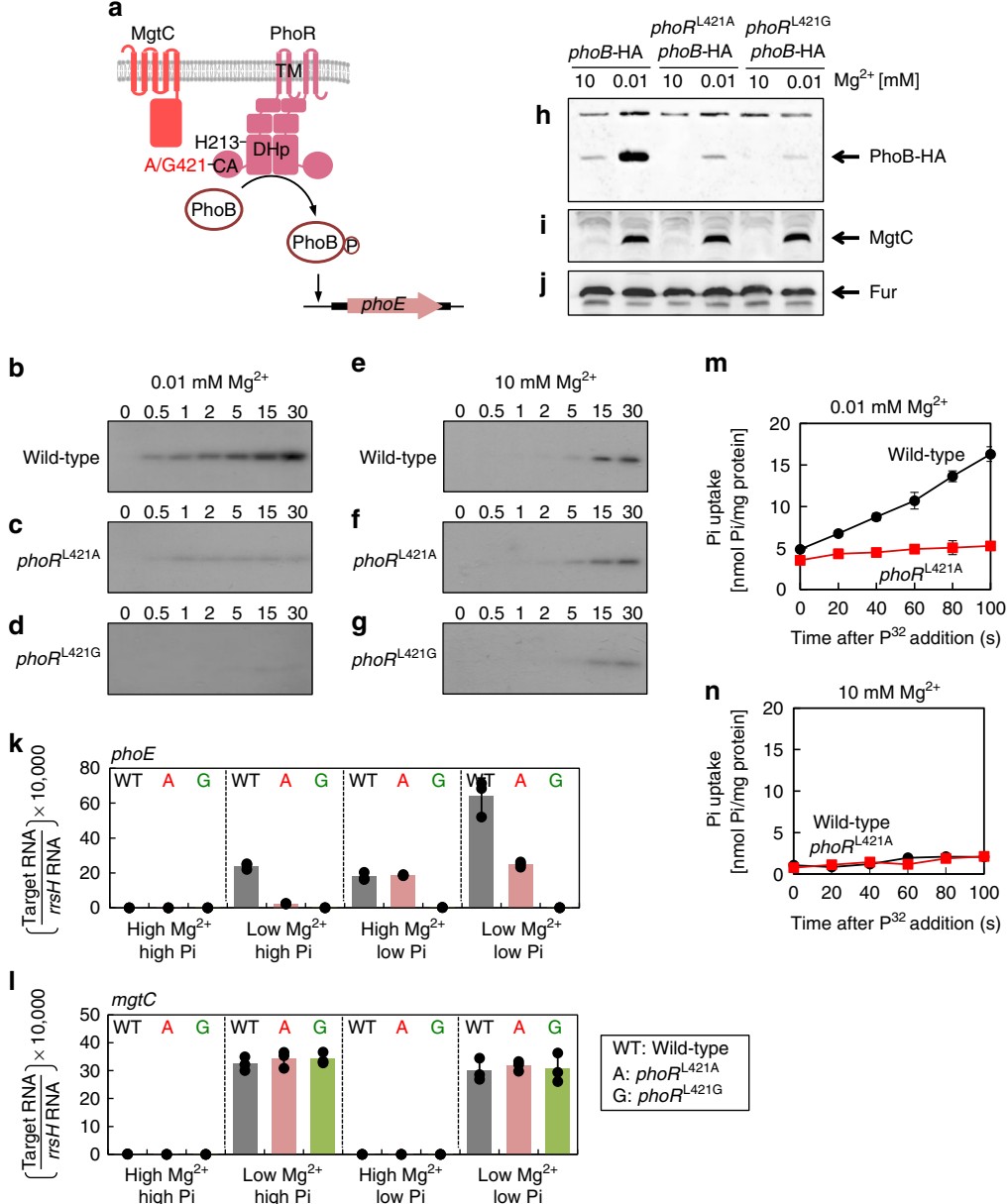

**Fig. 4** Leu421 to Ala or Gly substitution in PhoR prevents MgtC-mediated control of PhoR. **a** Schematic representation of the MgtC and PhoR histidine kinase. Ala or Gly-substituted 421 and His213 residues are indicated. **b**–**g** Autophosphorylation assay. Levels of PhoR-P following incubation of membrane vesicles prepared from *Salmonella* strains with the wild-type *phoR* (14028s, **b** and **e**), Leu421 to Ala-substituted *phoR* (EN949, **c** and **f**), or Leu421 to Gly-substituted *phoR* (EN991, **d** and **g**) genes grown for 5 h in N-minimal media containing 0.01 mM (**b**–**d**) or 10 mM (**e**–**g**) Mg$^{2+}$ at the indicated times. **h**–**j** Western blot analysis of crude extracts prepared from a chromosomal *phoB*-HA strain with the wild-type *phoR* gene (EN839), *phoR* derivative with the Leu 421 to Ala substitution (EN966), or *phoR* derivative with the Leu 421 to Gly substitution (EN1003) grown for 5 h in N-minimal media containing 0.01 mM or 10 mM Mg$^{2+}$. Blots were probed with anti-HA (**h**), anti-MgtC (**i**), or anti-Fur (**j**) antibodies to detect PhoB, MgtC, and Fur proteins, respectively. **k**, **l** Low Mg$^{2+}$ and low-phosphate signals increase *phoE* mRNA levels in an additive and independent manner. Relative mRNA levels of the *phoE* (**k**) and *mgtC* (**l**) genes in *Salmonella* strains with the wild-type *phoR* gene (WT, black), Leu421 to Ala-substituted *phoR* (A, red), or Leu421 to Gly-substituted *phoR* (G, green) genes. Bacteria were grown for 5 h in N-minimal media containing combinations of 10 mM (high Mg$^{2+}$) or 0.01 mM (low Mg$^{2+}$) Mg$^{2+}$ and 10 mM (high Pi) or 0.01 mM (low Pi) Pi. Data are represented as mean ± SD ($n = 3$ independent measurements). Relative mRNA levels represent (target RNA/ *rrsH* RNA) ×10,000. See also Supplementary Fig. 6. **m**, **n** Phosphate transport assay. Uptake of [$^{32}$P] orthophosphate into whole cells of *Salmonella* strains with the wild-type *phoR* (wild-type, black) or *phoR* $^{Leu421Ala}$ (*phoR* $^{L421A}$, red) gene grown in N-minimal media containing 0.01 mM (**m**) or 10 mM (**n**) Mg$^{2+}$. Levels of radioactive orthophosphate accumulated in cells were determined over time by liquid scintillation counting as described in Methods ($n = 3$ independent measurements). See also Supplementary Fig. 7

*a* subunit of the F$_1$F$_o$ ATP synthase and the *atpB mgtC* double mutant showed similar defects in intramacrophage survival[14]. In contrast, a *Salmonella* strain with the *phoR*$^{L421A}$ substitution in the *mgtC* mutant background showed a defect similarly to the *mgtC* deletion mutant (Fig. 5d) but differently to the mutant strains with the *phoR*$^{L421A}$ substitution or *phoB* deletion. Therefore, MgtC's effect on intramacrophage survival via its inhibitory interaction with the F$_1$F$_o$ ATP synthase appears to be dominant over its effect via its stimulatory interaction with PhoR.

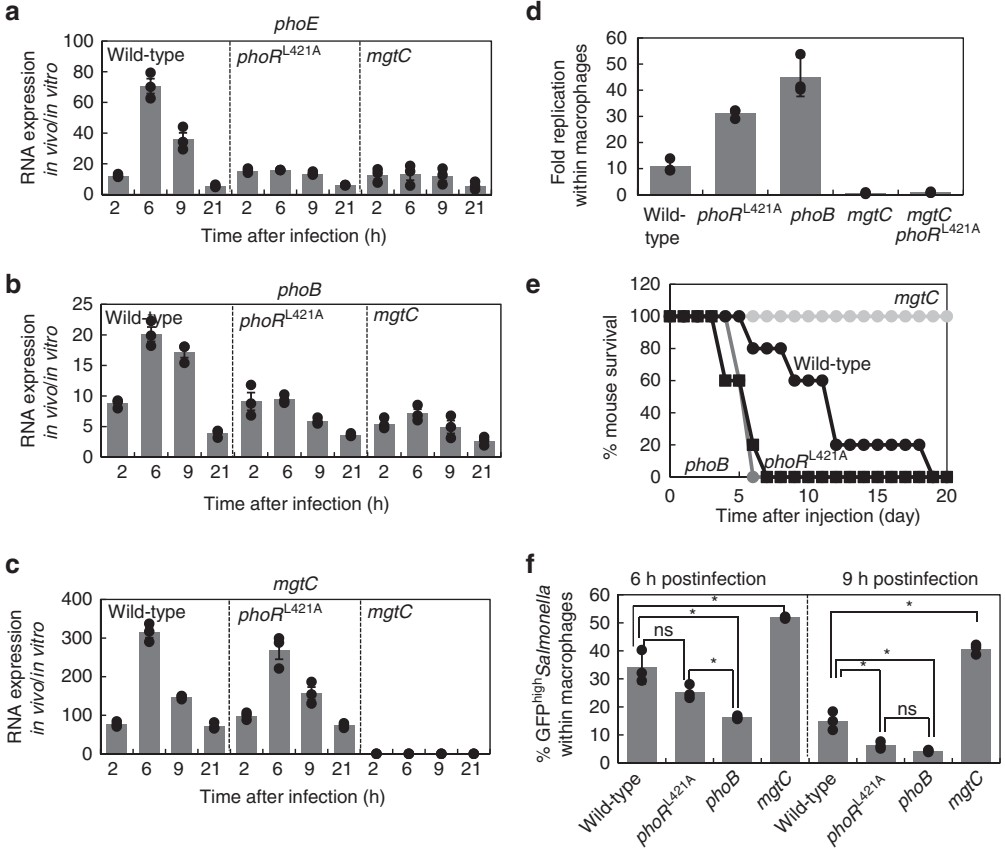

**Fig. 5** Leu421 to Ala substitution in PhoR promotes virulence in mice and decreases non-replicating *Salmonella* inside macrophages. **a–c** Relative mRNA levels of the *phoE* (**a**), *phoB* (**b**), and *mgtC* (**c**) genes produced by wild-type *Salmonella* (14028s), the *phoR* chromosomal mutant with Leu421 replaced by Ala codon (EN949), and an *mgtC* deletion mutant (EL4) inside J774 A.1 macrophages at the indicated times after infection (mean ± SEM, $n = 3$ independent infections). **d** Survival inside J774 A.1 macrophages of the *Salmonella* strains listed in **a–c**, the strain deleted for the *phoB* gene (KK10), and a strain with both the *phoR* Leu421Ala substitution and *mgtC* deletion (SM085) at 18 h post infection (T18). Fold replication represents [number of bacteria at T18/number of bacteria at T1]. Shown are the means and SD from three independent infections. **e** Survival of C3H/HeN mice inoculated intraperitoneally with ~$10^3$ colony-forming units of the *Salmonella* strains listed above. **f** Fluorescence dilution assay of the *Salmonella* strains listed in **e** inside J774 A.1 macrophages. The percentage of cells expressing high levels of GFP (GFP$^{High}$) was calculated from flow cytometric detection of mCherry and GFP fluorescence in *Salmonella* harboring pFCcGi plasmid ($n = 30,000$ cells). Shown are the means and SD from three independent infections. *$P < 0.05$, n.s. not significant, two-tailed *t*-test. See also Supplementary Fig. 8

Similar to what we observed in intramacrophage survival, the *phoR*$^{L421A}$ substitution or *phoB* deletion rendered *Salmonella* hypervirulent when mice were injected with ~$10^3$ CFUs of *Salmonella* intraperitoneally (Fig. 5e). Collectively, this section of data demonstrates that MgtC's regulatory action on expression of Pho genes via the PhoB/PhoR two-component system compromises *Salmonella*'s ability to survive within macrophages and virulence in mice.

**phoR$^{L421A}$ decreases non-replicating *Salmonella* inside host.** We identified that MgtC activates phosphate uptake by stimulating PhoR autophosphorylation (Fig. 3), and subsequently activating transcription of Pho genes (Fig. 4). This MgtC-mediated control of PhoR histidine kinase contributes to upregulation of mRNA levels of Pho genes inside macrophages (Fig. 5). Interestingly, the fact that the removal of MgtC-mediated activation of Pho genes by the *phoR*$^{L421A}$ substitution promotes *Salmonella* virulence indicates that wild-type *Salmonella* activates phosphate transport despite the fact that this property compromises *Salmonella*'s ability to replicate within macrophages and to cause disease in mice.

If this is the case, then what would be the benefit(s) of such an increase in phosphate uptake during *Salmonella* infection? We

reasoned that the increase in phosphate uptake mediated by the MgtC virulence protein might impact on the formation of non-growing/slow-growing *Salmonella* inside macrophages because the increase in phosphate uptake apparently reduces replication inside macrophages (Fig. 5d), and also because the deletion of a gene involved in the phosphate signaling pathway has been reported to be involved in the formation of persisters[32–34], which are phenotypically similar to non-replicating cells.

To measure the formation of non-replicating *Salmonella*, we used a dual fluorescence-based assay[35] by introducing a plasmid expressing GFP from an arabinose-inducible promoter and mCherry from a constitutive promoter (Supplementary Fig. 8). In principle, because GFP- and mCherry-expressing *Salmonella* infected macrophages without arabinose, replicating *Salmonella* would dilute the GFP fluorophore, leading to a decrease in GFP levels as they divide, whereas non-replicating *Salmonella* would retain high GFP levels (Supplementary Fig. 8). Wild-type *Salmonella* retained 33.9% and 15.0% of GFP-high cells inside macrophages at 6 and 9 h post infection, respectively (Fig. 5f). The occurrence of GFP-high cells is partly due to *Salmonella*'s ability to transport phosphate via MgtC inside macrophages, because *Salmonella* strains lacking MgtC's binding to PhoR or those lacking the PhoB response regulator decreased percentages

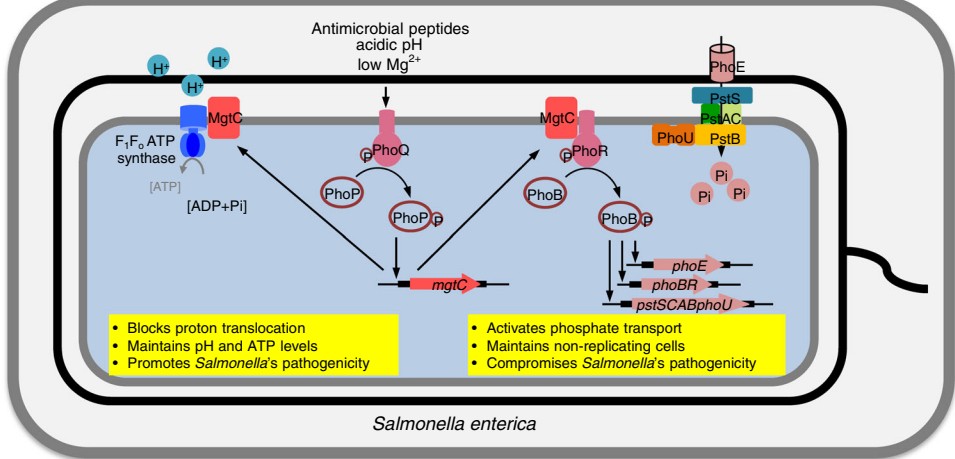

**Fig. 6** MgtC reprograms *Salmonella*'s ion uptake inside a macrophage phagosome. On the one hand, MgtC binds to $F_1F_0$ ATP synthase and blocks proton movement across the bacterial inner membrane. This interaction allows *Salmonella* to maintain cytoplasmic pH and ATP levels and promote pathogenicity. On the other hand, MgtC binds to PhoR histidine kinase and activates phosphate transport in the bacterial membrane by promoting transcription of the PhoB-dependent genes involved in phosphate transport. Phosphate uptake mediated by MgtC is involved in the maintenance of non-replicating *Salmonella* inside macrophages

of non-replicating cells at 9 h post infection (6.2% and 4.3%, respectively) compared to those detected at 6 h post infection (25.2% and 16.1%, respectively)(Fig. 5f). These data suggest that MgtC-mediated phosphate uptake is involved in the formation of a non-replicating *Salmonella* population during infection. The *mgtC* mutant exhibited an increased fraction of GFP-high cells at 6 and 9 h post infection (Fig. 5f), implying that MgtC might have multiple targets affecting the formation of non-replicating cells.

## Discussion
We have established that the MgtC virulence protein targets the PhoR histidine kinase, leading to activation of genes involved in phosphate transport and an increase in the non-replicating *Salmonella* population during infection. MgtC binds to the CA domain of PhoR histidine kinase (Figs. 2 and 3), thereby activating PhoR autophosphorylation to promote transcription of Pho genes (Figs. 3 and 4). MgtC's action toward PhoR allows *Salmonella* to ensure phosphate uptake independent of the presence of a cognate signal, which might not be guaranteed during infection. Phosphate acquisition seems to be critical for normal *Salmonella* virulence because the *phoR* substitution that prevented MgtC's binding lost the ability to activate Pho regulon during infection and rendered *Salmonella* hypervirulent (Fig. 5). These findings reveal an unexpected molecular link between two previously observed phenomena: the increases in the mRNA levels of the *mgtC* gene and *pstSCABphoU* operon during *Salmonella* infection[18,36,37].

It was proposed that MgtC activates PhoB indirectly by decreasing cytoplasmic Pi levels via a decrease in translation that limits the release of free cytoplasmic Pi from ATP[36]. This conclusion was drawn from the data that the strain lacking MgtC, which was previously shown to destabilize ribosome assembly in low $Mg^{2+}$ (ref. [38]), accumulated steady-state levels of intracellular phosphate when cells were grown for 5 h in low phosphate and low $Mg^{2+}$[36]. However, we could not detect the elevated phosphate levels of the *mgtC* mutant when we tested our strains in the same media (Supplementary Fig. 9). In addition, our findings unequivocally demonstrate that: (i) MgtC activates mRNA levels of the PhoB-dependent genes directly by interacting with PhoR histidine kinase and (ii) MgtC exerts its effects on the increase in

PhoR autophosphorylation, mRNA levels of the PhoB-dependent genes, and phosphate uptake even when expressed from the IPTG-inducible promoter for 1 h without altering intracellular phosphate levels (Supplementary Fig. 10). Given that MgtC overexpression does increase radioactive phosphate uptake immediately (Supplementary Fig. 7) but does not alter the intracellular phosphate levels in high-phosphate medium (Supplementary Fig. 10), this result reflects that steady-state levels of intracellular phosphate are relatively high and less affected by the amount of Pi transported via MgtC in this condition. Cumulatively, these data argue that MgtC induces mRNA levels of the PhoB-dependent genes independent of cytoplasmic Pi levels.

PhoQ senses low $Mg^{2+}$, acidic pH, or antimicrobial peptides[20–22] that could be encountered within a macrophage phagosome and phosphorylates PhoP to promote transcription of the PhoP-dependent genes, including *mgtC* (Fig. 6)[19,39]. Likewise, PhoR senses low phosphate and activates PhoB via a phosphorelay to promote transcription of Pho genes, including *phoE* and *pstSCAB*[1,2]. MgtC allows *Salmonella* to promote transcription of Pho genes in response to the PhoP/PhoQ-activating signals, low $Mg^{2+}$, in addition to the PhoB/PhoR-activating signal, low phosphate. The PhoP/PhoQ- and PhoB/PhoR-activating signals exert their effects on transcription of the Pho genes independently and additively because *Salmonella* grown in both low $Mg^{2+}$ and low phosphate further increased *phoE* mRNA levels, compared to those grown in low $Mg^{2+}$ or low phosphate alone (Fig. 4). However, PhoP-dependent transcription is not influenced by the PhoB/PhoR-activating signal because low phosphate does not affect *mgtC* transcription (Fig. 4). MgtC's function that connects the PhoP/PhoQ and PhoB/PhoR two-component systems is reminiscent of that performed by the *Salmonella* PmrD protein. PmrD is produced by the PhoP/PhoQ two-component system in response to low $Mg^{2+}$ and then binds to the PmrB histidine kinase, thereby promoting transcription of the PmrA-dependent genes involved in lipopolysaccharide (LPS) modification/resistance to antimicrobial peptides[40,41]. As PmrB normally responds to high $Fe^{3+}$ (ref. [42]), PmrD enables *Salmonella* to transcribe the PmrA-regulated genes in response to two different signals: low $Mg^{2+}$ and high $Fe^{3+}$. MgtC and PmrD are similar to each other in a sense that the post transcriptional regulation occurred by MgtC and PmrD allows *Salmonella* to produce phosphate uptake proteins and LPS-modifying

enzymes in low $Mg^{2+}$, even in the absence of cognate signals for the second two-component systems. However, the ways in which these proteins achieve activation of the second two-component system clearly differ because MgtC binds to the PhoR histidine kinase and activates PhoB-dependent transcription by promoting PhoR autophosphorylation activity, whereas PmrD binds to the PmrB histidine kinase and activates PmrA-dependent transcription by inhibiting PmrB phosphatase activity towards phospho-PmrA[43]. In addition, MgtC is an inner membrane protein, whereas PmrD is a small cytoplasmic protein, which belongs to a group called connector/adaptor proteins[44,45].

One way to understand the signal(s) or environment during infection is to identify bacterial genes specifically expressed during infection. Previous large-scale transcriptome analyses or promoter-trapping techniques searching for promoters/genes induced inside host[18,30,46,47] have identified that many genes are highly transcribed inside macrophages or during mouse infection. These identified genes allowed us to reasonably deduce that the environment of the *Salmonella*-containing phagosome would be limiting in ions, poor in nutrients, and rich in antimicrobial compounds[18]. It has been proposed that the environment *Salmonella* experiences during infection might be limiting in phosphate ions[18]. This assumption is based on the fact that mRNA levels of the *pstSCAB* genes are highly induced when *Salmonella* is inside macrophages[18,48] and that the PhoB/PhoR two-component system controlling transcription of the *pstSCAB* genes is normally turned on in low phosphate[1]. However, the physiological relevance of low phosphate within a macrophage phagosome is presently unclear because elevated mRNA levels of Pho regulon during infection are actually mediated by MgtC's regulatory action. In fact, we identified the *phoR*[L421A] mutant that is unable to be activated by MgtC but still responds to low phosphate (Fig. 4 and Supplementary Fig. 6). Surprisingly, this *phoR* substitution mutant did not induce mRNA expression of the PhoB-dependent genes, *phoE* and *phoB*, at 6 and 9 h post infection and behaved similarly to the *mgtC* mutant (Fig. 5), arguing against the notion that the phagosomal environment during *Salmonella* infection is phosphate-limiting.

Unexpectedly, the *phoR* mutant that prevents MgtC binding and thus loses the ability to transport phosphate in the presence of MgtC (Fig. 4 and Supplementary Fig. 7) promotes *Salmonella*'s replication inside macrophages and virulence in mice (Fig. 5). In other words, wild-type *Salmonella* promotes phosphate uptake via MgtC despite decreasing replication efficiency within macrophages and virulence in mice (Fig. 5). In turn, this property contributes to establishing a non-replicating *Salmonella* population inside macrophages (Fig. 5). The ability to maintain the non-replicating population is associated with the ability to transport phosphate because both the strain lacking the PhoB response regulator and the strain preventing MgtC-mediated PhoR activation decrease non-replicating *Salmonella* inside macrophages at 9 h post infection (Fig. 5). Phosphate limitation induces expression of the phosphate-specific transporter operon and triggers the formation of non-replicating bacilli in *Mycobacterium tuberculosis*[49,50]. This is interesting because, although *Mycobacterium* harbors the *mgtC* gene that is also required for intramacrophage survival and virulence in mice[10], it has not yet been addressed whether a molecular connection exists between MgtC and low phosphate-induced non-replicating bacilli in this organism. Given that *Salmonella* requires the *mgtC* gene for long-term systemic infection[51], our findings highlight the critical role that phosphate metabolism might play in pathogens' persistence.

We previously identified that MgtC targets the *a* subunit of the $F_1F_o$ ATP synthase[14]. As MgtC binds to the membrane-bound fractions of the $F_1F_o$ ATP synthase, it blocks proton translocation across the inner membrane and thus inhibits proton transport-coupled ATP synthesis (Fig. 6). Here, we establish that MgtC targets to the PhoR histidine kinase. In this case, MgtC's binding to PhoR promotes phosphate transport across the inner membrane by activating transcription of the PhoB-dependent genes (Fig. 4 and Supplementary Fig. 7). These findings reveal that the MgtC virulence protein reprograms *Salmonella*'s ion uptake (proton and phosphate) inside a macrophage phagosome (Fig. 6). Interetingly, MgtC's action on the $F_1F_o$ ATP synthase within an acidified phagosome results in a decrease in ATP levels and membrane potential of the bacterium. The decrease in ATP levels promotes intramacrophage survival by decreasing biofilm formation[52], which might not be beneficial for *Salmonella* to replicate within macrophages. By contrast, MgtC's action on the PhoR histidine kinase increases phosphate uptake and compromises *Salmonella*'s survival inside macrophages, possibly by increasing a population of non-replicating *Salmonella* (Fig. 6). These findings illustrate how a single virulence protein manipulates a bacterium's physiological status to modulate intramacrophage survival differently by targeting multiple proteins.

## Methods

**Bacterial strains, plasmids, primers, and growth conditions**. The bacterial strains and plasmids used in this study are listed in Supplementary Table 1. All *Salmonella enterica* serovar Typhimurium strains are derived from the wild-type strain 14028s[53] and were constructed by phage P22-mediated transductions[54]. All DNA oligonucleotides are listed in Supplementary Table 2. Bacteria were grown at 37 °C in Luria-Bertani broth (LB), and N-minimal media (pH 7.7)[55] supplemented with 0.1% casamino acids, 38 mM glycerol, and the indicated concentrations of $MgCl_2$. In case of low-phosphate N-minimal media, 10 mM $KH_2PO_4$ in the N-minimal media was replaced by 0.01 mM $KH_2PO_4$. *Escherichia coli* DH5α was used as the host for the preparation of plasmid DNA, and BTH101 lacking the *cya* gene was used as the host for the bacterial two-hybrid system[56]. For Supplementary Fig. 9, we used modified MOPS media lacking $CaCl_2$ and containing 0.5 mM $Mg^{2+}$ [36]. Ampicillin was used at 50 μg ml$^{-1}$, chloramphenicol at 25 μg ml$^{-1}$, kanamycin at 50 μg ml$^{-1}$, or tetracycline at 10 μg ml$^{-1}$. IPTG (isopropyl β-D-1-thiogalactopyranoside) was used at 0.25 mM, L-arabinose at 1 mM, and X-Gal (5-bromo-4-chloro-3-indolyl β-D-galactopyranoside) at 80 μM.

**RNA sequencing to identify genes affected by *mgtC* expression**. Bacteria were grown overnight in N-minimal medium containing 10 mM $Mg^{2+}$. A 1/100 dilution of the overnight culture was used to inoculate 10 ml of the same medium, and grown for 3 h. Cells were then washed and transferred to 10 ml of N-minimal medium containing 0.5 mM $Mg^{2+}$ and 0.25 mM IPTG to induce *mgtC* expression, and then grown for an additional hour. Bacteria were harvested and RNA was isolated for further analysis. Total RNA was prepared by using RNeasy mini kit (Qiagen) and the integrity of the RNA samples was measured by using BioAnalyzer 2100 (Agilent Technologies). The samples with an RNA integrity number of over 8.0 were used for the next step. For ribosomal RNA depletion, 5 μg of the total RNA was processed by Ribo-Zero rRNA Removal Kit (Bacteria) (Illumina, MRZMB126). Sequencing libraries for RNA-Seq were constructed using TruSeq Stranded Total RNA Library Prep Kit (Illumina, RS-122-2201), following the manufacturer's instructions. Final sequencing libraries were quantified by Pico-Green assay (Life Technologies) and visualized using BioAnalyzer 2100. Sequencing was performed at ChunLab Inc. (Korea) using a HiSeq 2500 and NextSeq 500 instrument (Illumina Inc.), following the manufacturer's protocol, which generated 100 bp single-end reads (HiSeq 2500) and 75 bp paired-end reads (NextSeq 500) for each sample. The sequencing adapter removal and quality-based trimming for the raw data were performed using Trimmomatic v. 0.36[57] with TruSeq adapter sequences. Cleaned reads were mapped to the reference genome using bowtie2[58] with a default parameter. For counting the reads mapped to each CDS (coding sequence), featureCounts[59] was used. Finally, the count from each CDS was normalized to FPKM (fragments per kilobase million) and TPM (transcripts per million)[60] value. Processed data were deposited in the Gene Expression Omnibus (GEO) database with accession number GSE103153.

**Plasmid construction**. pTGFP-*phoR* and pTGFP-*phoR*[Leu 421 Ala] plasmids expressing the C-terminally GFP-fused PhoR and PhoR[Leu 421 Ala] proteins from the constitutive p*lac* promoters were constructed as follows. The *phoR* and *phoR*[Leu 421 Ala] genes were amplified by PCR using primers KHU450 and KHU451 and 14028s or EN949 (*phoR*[Leu 421 Ala]) genomic DNA as templates. The PCR products were purified and introduced between *Eco*RI and *Bam*HI restriction sites of pTGFP[25].

For the bacterial two-hybrid assays, the plasmids pUT18-*mgtC*, pUT18-*mgtC* 130–231, pKT25-*phoR*, pKT25-*phoU*, pKT25-*pstA*, pKT25-*pstB*, pKT25-*phoR_TM_ domain*, pTK25-*phoR_PAS_domain*, pKT25-*phoR_CA_domain*, pKT25-*phoR_TM*

+*PAS_domains*, pKT25-*phoR 1–417*, pKT25-*phoR 1–418*, pKT25-*phoR 1–419*, pKT25-*phoR 1–420*, pKT25-*phoR 1–421*, pKT25-*phoR 1-422*, pKT25-*phoR 1-423*, pKT25-*phoR 1-424*, pKT25-*phoR 1-425*, pKT25-*phoR 1-426*, pKT25-*phoR* Leu 421 Ala, pKT25-*phoR* Leu 421 Ile, pKT25-*phoR* Leu 421 Val, and pKT25-*phoR* Leu 421 Gly were constructed as follows: DNA fragments corresponding to the *mgtC*, *mgtC 130–231*, *phoR*, *phoU*, *pstA*, *pstB*, *phoR_TM_domain*, *phoR_PAS_domain*, *phoR_CA_domain*, *phoR_TM+PAS_domains*, *phoR 1-417*, *phoR 1-418*, *phoR 1-419*, *phoR 1-420*, *phoR 1-421*, *phoR 1-422*, *phoR 1-423*, *phoR 1-424*, *phoR 1-425*, *phoR 1-426*, *phoR* Leu 421 Ala, *phoR* Leu 421 Ile, *phoR* Leu 421 Val, and *phoR* Leu 421 Gly were amplified by PCR using the primer pairs, KHU847/KHU848 (for *mgtC*), KHU343/KHU344 (for *mgtC 130–231*), KHU153/KHU154 (for *phoR*), KHU155/KHU156 (for *phoU*), KHU157/KHU158 (for *pstA*), KHU159/KHU160 (for *pstB*), KHU187/KHU188 (for *phoR_TM_domain*), KHU189/KHU190 (for *phoR_PAS_domain*), KHU191/KHU192 (for *phoR_CA_domain*), KHU187/KHU190 (for *phoR_TM +PAS_domains*), KHU363/KHU541 (for *phoR 1-417*), KHU363/KHU540 (for *phoR 1-418*), KHU363/KHU539 (for *phoR 1-419*), KHU363/KHU489 (for *phoR 420*), KHU363/KHU490 (for *phoR 1-421*), KHU363/KHU491 (for *phoR 1-422*), KHU363/KHU492 (for *phoR 1-423*), KHU363/KHU493 (for *phoR 1-424*), KHU363/KHU511 (for *phoR 1-425*), KHU363/KHU463 (for *phoR 1-426*), KHU363/KHU542 (for *phoR* Leu421Gly), KHU363/KHU543 (for *phoR* Leu421Ala), KHU363/KHU544 (for *phoR* Leu421Val), and KHU363/KHU545 (for *phoR* Leu421Ile), using 14028s genomic DNA as a template. For pUT18-*mgtC* Asn 92 Thr, DNA fragments were amplified by PCR using the primer pairs KHU847/KHU848 and EL551 genomic DNA as a template. After purification, the PCR products were digested with *Bam*HI and *Kpn*I and introduced between the *Bam*HI and *Kpn*I sites of the plasmid vector pKT25[61] or between the *Bam*HI and *Kpn*I sites of the pUT18 plasmid[61]. For pKT25-*mgtR*, KHU849 and KHU850 oligomers were annealed and ligated to the pKT25 plasmid digested with *Bam*HI and *Kpn*I restriction enzymes.

The plasmids pBAD33-*mgtC*, pBAD33-*mgtC*-FLAG, and pBAD33-*phoR*-HA were constructed as follows: PCR fragments corresponding to the *mgtC* and *phoR* genes were generated by PCR with the primers KHU131 and KHU132 (for *mgtC*), KHU881 and KHU882 (for *mgtC*-FLAG), KHU195 and KHU196 (for *phoR*), and 14028s genomic DNA as a template, digested with *Eco*RI and *Hin*dIII (for *mgtC*), or *Xba*I and *Hin*dIII (for *mgtC*-FLAG and *phoR*), and cloned into pBAD33 digested with the same enzymes. The sequences of the resulting constructs were verified by DNA sequencing.

**Construction of the *phoR*L421A and *phoR*L421G mutants.** To generate strains with chromosomal mutations in the *phoR* coding region, we used the fusaric acid-based counter selection method[27]. First, we introduced a Tet[R] cassette in the *phoR* gene as follows: we generated PCR products harboring the *tetRA* genes using the primers KHU355/KHU356 and MS7953s genomic DNA as a template. The product was purified using a QIAquick PCR purification kit (QIAGEN) and used to electroporate *Salmonella* 14028s containing plasmid pKD46[62]. The resulting *phoR*::*tetRA* (KK149) strain containing plasmid pKD46 was kept at 30 °C. Then, we replaced the *tetRA* cassettes by preparing DNA fragments carrying leucine to alanine or leucine to glycine codon substitutions in *phoR* at position 421. The DNA fragments with the Leu421 substitutions were prepared by a two-step PCR process. For the first PCR, we used two sets of primer pairs, KHU351/KHU597 and KHU596/KHU352 (for 421st leucine codon to alanine), and KHU351/KHU579 and KHU580/KHU352 (for 421st leucine codon to glycine), and 14028s genomic DNA as a template. For the second PCR, we mixed the two PCR products from the first PCR as templates and amplified the DNA fragments using the primers KHU351/KHU352. The resulting PCR products were purified and integrated into the KK149 chromosome and selected against tetracycline resistance with media containing fusaric acid[63] to generate EN949 (*phoR* Leu 421 Ala) and EN991 (*phoR* Leu 421 Gly), tetracycline-sensitive, ampicillin-sensitive (Tet[S] Amp[S]) chromosomal mutants. The presence of the expected nucleotide substitutions was verified by DNA sequencing.

**Construction of a *phoB* deletion mutant.** A *Salmonella* strain deleted for the *phoB* gene was generated by the one-step gene inactivation method[62]. A Km[R] cassette for the *phoB* gene was PCR amplified from plasmid pKD4 using primers DE-*phoB*-F/DE-*phoB*-R (for *phoB*), and the resulting PCR products were integrated into the 14028s chromosome to generate EN296 (*phoB*::Km[R]). The *phoB* strain (KK10) was generated by removing the Km[R] cassette from EN296 using the pCP20 plasmid[62].

**Construction of strains with the HA-tagged *phoB* gene.** A *Salmonella* strain with an HA-tag at the C-terminus of the *phoB* gene was generated by the one-step gene inactivation method[62]. Cm[R] cassettes were PCR amplified from the pKD3 plasmid, using the primers HKU435 and KHU436, and the resulting PCR products were integrated into the wild-type 14028s, *phoR*Leu421Ala (EN949), or *phoR*Leu421Gly (EN991) chromosomes to generate EN836 (*phoB*-HA::Cm[R]), EN964 (*phoR*-Leu421Ala, *phoB*-HA::Cm[R]), and EN1000 (*phoR*Leu421Gly, *phoB*-HA::Cm[R]), respectively. The *phoB*-HA (EN839), *phoR*Leu421Ala *phoB*-HA (EN966), and *phoR*Leu421Gly *phoB*-HA (EN1003) strains were constructed by removing the Cm[R] cassettes from EN842, EN964, and EN1000 using the pCP20 plasmid[62].

**Bacterial two-hybrid (BACTH) assay.** To assess protein (or peptide)–protein interactions in vivo, a BACTH assay was conducted[56]. The *Escherichia coli* BTH101 (*cya*) strain was co-transformed with derivatives of the pUT18 and pKT25 plasmids. The strains were grown overnight at 30 °C in LB supplemented with ampicillin (50 μg ml$^{-1}$) and kanamycin (50 μg ml$^{-1}$). Then, 2 μl of cells was spotted on solid LB medium containing 100 μM IPTG, 100 μM ampicillin, 100 μM kanamycin, and 80 μM X-Gal, followed by incubation at 30 °C for 40 h. For more detailed quantitative analysis, β-galactosidase assays were performed[64].

**Protein structure modeling and protein docking modeling.** We used Protein Homology/analogY Recognition Engine V 2.0 (Phyre2)[65] to model the structure of the cytoplasmic portion of the PhoR protein (corresponding to amino acids 52–431) from *Salmonella enterica* serovar Typhimurium 14028s. The structure of the *Salmonella* PhoR protein was modeled based on the structure of VicK from *Streptococcus mutans* (PDB: 4I5S)[66]. Then, we used the ClusPro webserver[67,68] to dock a dimer structure of the *Salmonella* PhoR protein.

**Western blot analysis.** Cells were grown for 5 h in 15 ml of N-minimal medium containing 10 mM or 0.01 mM Mg$^{2+}$. Cells were normalized by measuring the optical density of the culture medium at a wavelength of 600 nm (OD$_{600}$). Crude extracts were prepared in TBS (Tris-buffered saline) buffer by sonication. Whole-cell lysates were resolved on 12% sodium dodecyl sulfate (SDS)-polyacrylamide gels, and the separated proteins were transferred onto nitrocellulose membranes and incubated with monoclonal anti-HA antibodies (Santa Cruz, 1:10,000 dilution, sc-805) overnight. The blots were developed by incubation with anti-rabbit IgG horseradish peroxidase-linked antibody (ThermoFisher, 1:10,000 dilution, 31460) for 1 h, and were visualized using the ECL detection system (SuperSignal® West Femto Maximum Sensitivity Substrate, Thermo). The unprecessed scans of blots with the locations of molecular weight markers are provided in the Source Data file.

**Immunoprecipitation assay.** The interaction between the MgtC and PhoR proteins was investigated in wild-type *Salmonella* expressing the *mgtC* gene from an arabinose-inducible plasmid (pBAD33-*mgtC* or pBAD33-*mgtC*-FLAG) and C-terminally *gfp*-tagged *phoR* gene, its derivatives (pTGFP-*phoR*, pTGFP-*phoR* Leu421Ala, or pTGFP-*phoR* Leu421Gly), or *Salmonella* expressing the empty vector (pTGFP) from the constitutive p$_{lac}$ promoter[25]. Cells were grown overnight in N-minimal media containing 10 mM Mg$^{2+}$. A 1/100 dilution of the bacterial culture was inoculated in 15 ml of N-minimal media containing 10 mM Mg$^{2+}$, and grown for 3 h. Cells were then washed and transferred to 15 ml of N-minimal media containing 0.5 mM Mg$^{2+}$ and 1 mM L-arabinose, and grown for 1 h. Cells were normalized by measuring OD$_{600}$. Crude extracts were prepared in TBS (Tris-buffered saline) buffer by sonication. For a pull-down assay with anti-GFP antibodies, 50 μl of the crude extracts were kept for input and 500 μl of the protein extracts were mixed with 25 μl of GFP-Trap®_A beads (Chromotek) for 1 h at 4 °C on nutator (BenchMark), according to the manufacturer's instructions. After washing the beads, the bound proteins were eluted in SDS sample buffer, separated on a 12% SDS-polyacrylamide gel, and analyzed by western blotting using anti-MgtC (1:10,000 dilution, polyclonal antibodies raised against purified MgtC proteins) and anti-GFP (1:10,000 dilution, Rockland™, 600-401-215) antibodies for 2 h. For a pull-down assay with anti-FLAG antibodies, 50 μl of the crude extracts were kept for input and 500 μl of the protein extracts were mixed with 25 μl of EZview™ Red anti-FLAG® M2 affinity gel (Sigma-Aldrich, F2426) for 1 h at 4 °C on nutator (BenchMark), according to the manufacturer's instructions. After washing the beads, bound proteins were eluted in SDS sample buffer and separated on 12% SDS-polyacrylamide gel and analyzed by western blotting using anti-FLAG (1:10,000 dilution, Sigma-Aldrich, F7425) and anti-GFP (1:10,000 dilution, Rockland™, 600-401-215) antibodies for 2 h. The blots were washed and hybridized with anti-rabbit IgG horseradish peroxidase-linked antibodies (1:20,000 dilution, ThermoFisher, 31460) for 1 h and detected using SuperSignal® West Femto Maximum Sensitivity Substrate (ThermoFisher).

**Membrane vesicle preparation.** Cells were grown for 5 h in 15 ml of N-minimal medium containing 10 mM or 0.01 mM Mg$^{2+}$ and 10 mM or 0.01 mM Pi. Cells were normalized by measuring OD$_{600}$. Crude extracts were prepared in TBS (Tris-buffered saline) buffer by sonication. After removing cell debris, the membranes were isolated by centrifugation for 2 h at 40,000 × $g$ at 4 °C (Optima™ TLX Ultracentrifuge, Type 90Ti Rotor, BeckmanCoulter). The pellets were resuspended in 500 μl of TBS buffer. The protein concentration was determined using a Nanodrop machine (ThermoFisher).

**Autophosphorylation assay.** Fifty microliters of membrane vesicles expressing PhoR, PhoR Leu 421 Ala, or PhoR Leu 421 Gly were incubated in 100 μl of TBS (Tris-buffered saline) containing 1 mM MgCl$_2$ at room temperature. The reaction was started with the addition of [γ$^{32}$-P] ATP (10 μCi, PerkinElmer) to the mixture[69]. A 10 μl aliquot was mixed with 10 μl of 5x SDS loading buffer (Biosesang) at different time points to stop the reaction. The samples were kept on ice until they were loaded onto a 12% SDS-polyacrylamide gel. After electrophoresis, the gel was dried on the membrane using Model 583 gel dryer (Bio-Rad) and then autoradiographed. Phosphorylated PhoR proteins were identified using samples prepared from wild-

type and the *phoR* mutant *Salmonella* grown for 5 h in N-minimal media containing 0.01 mM Pi, a PhoB/PhoR-inducing condition.

**Phosphate transport assay.** *Salmonella* strains with the wild-type *phoR* and *phoR* Leu 421 Ala were grown in N-minimal medium with 10 mM and 0.01 mM $Mg^{2+}$ for 5 h. Cells were normalized by measuring $OD_{600}$. Whole cells were washed in TBS (Tris-buffered saline) buffer and resuspended in 300 μl of TBS buffer. Then, 300 μl of the cells were incubated at room temperature, and the reaction was started by adding 25 μCi of radioactive Pi solution (Phosphorus-32 Radionuclide [$^{32}$P]; 2 mCi (74 MBq); PerkinElmer) diluted in 50 μl of distilled water. During this experiment, 50 μl of each sample was collected every 20 s, and the reactions were stopped by the addition of 500 μl of 0.1 M LiCl and rapid filtration through 0.45 μm PROTRAN BA 85 membrane filters (Whatman$^{TM}$) with an applied vacuum. The filters were washed twice with 2 ml of 0.1 M LiCl, and then air-dried in 20 ml scintillation vials and soaked in 4 ml of scintillation fluid (HIDEX). The amount of radioactivity taken up by the cells was determined with a scintillation counter (Triathler multilabel tester, HIDEX) using the $^{32}$P-window and by counting each vial for 10 s. The amount of Pi taken up by the cells was calculated from the counts on the filters at each time-point relative to a control without cells, and data were plotted as specific activity [nmol Pi per mg protein].

**Measuring intracellular phosphate levels.** *Salmonella* strains were grown for 5 h in N-minimal medium with 10 mM and 0.01 mM $Mg^{2+}$. *Salmonella* strains harboring a plasmid with the *mgtC* gene were grown for 3 h in N-minimal medium with 10 mM $Mg^{2+}$, and then for an additional 1 h in the same media containing 0.5 mM $Mg^{2+}$ and 0.25 mM IPTG. Cells were normalized by measuring $OD_{600}$, washed twice with TBS (Tris-buffered saline) buffer, and resuspended in 400 μl of TBS buffer. Crude extracts were prepared in TBS (Tris-buffered saline) buffer by sonication. Then, 10 μl of the crude extracts were aliquoted into a 96-well plate and mixed with 140 μl of EnzChek Phosphate Assay (Thermo Fischer, E-6646) reagents and incubated for 30 min at 22 °C. Phosphate levels were determined by measuring the absorbance of the solutions at 360 nm using Synergy H1 plate reader (BioTek). When we measured intracellular phosphate levels of cells grown in modified MOPS media containing 0.5 mM Pi, we used molybdenum blue method[36] and measured the absorbance at 820 nm using Synergy H1 plate reader (BioTek).

**Measuring intracellular ATP levels.** For measuring the intracellular ATP levels, we used the BacTiter-Glo$^{TM}$ Microbial Cell Viability Assay (Promega, G8230), according to the manufacturer's instruction with slight modifications. Briefly, bacteria were grown overnight in N-minimal media containing 10 mM $Mg^{2+}$. Then, 50 μl of the overnight culture was washed in N-minimal media without $Mg^{2+}$ and grown for 5 h in 5 ml of N-minimal media containing 0.01 mM or 10 mM $Mg^{2+}$. Cells were normalized by measuring $OD_{600}$ and resuspended in 1 ml of PBS (phosphate-buffered saline). Then, 80 μl of this cell suspension was dispensed into a opaque 96-well microplate (PerkinElmer), followed by the addition of 80 μl of BacTiter-Glo$^{TM}$ Reagent. The contents were then mixed briefly by pipetting and incubated for 5 min. The luminescence of the samples was measured using Synergy H1 plate reader (BioTek).

**Quantitative real-time polymerase chain reaction (qRT-PCR).** Total RNA was isolated using RNeasy Kit (QIAGEN), according to the manufacturer's instructions. The purified RNA was quantified using a Nanodrop machine (NanoDrop Technologies). Complementary DNA (cDNA) was synthesized using Prime-Script$^{TM}$ RT reagent Kit (TaKaRa). The mRNA levels of the *mgtC* and *phoE* genes were measured by quantifying the cDNA using SYBR Green PCR Master Mix (TOYOBO) and the appropriate primers (*mgtC*: 7530/7531, *phoE*: KHQ015/ KHQ016, *pstS*: Q-pstS-F/Q-pstS-R, *phnS*: Q-phnS-F/Q-phnS-R, *ugpB*: Q-ugpB-F/ Q-ugpB-R, *phoB*: Q-phoB-F/Q-phoB-R), and monitored using a 7300 Real-Time PCR system (Applied Biosystems, Foster City) or StepOnePlus Real-Time PCR system (for Fig. 4 and Supplementary Figs. 1, 2, and 6). The mRNA levels of each target gene were calculated using a standard curve of the 14028s genomic DNA, and the data were normalized to the levels of 16S ribosomal RNA amplified with the primers 6970 and 6971.

**Macrophage survival assay.** Intramacrophage survival assays were conducted using the macrophage-like cell line J774 A.1[31]. Briefly, $5 \times 10^5$ macrophages in Dulbecco's modified Eagle's medium (DMEM) supplemented with 10% fetal bovine serum (FBS) were seeded in 24-well plates and cultured at 37 °C. Overnight grown bacteria were added to the macrophages at a multiplicity of infection (MOI) of 10:1, and the plates were centrifuged at 1000 rpm for 5 min at room temperature and incubated for an additional 20 min. Then, the extracellular bacteria were washed three times with PBS (phosphate-buffered saline) and killed by incubation with DMEM supplemented with 10% FBS and 120 μg ml$^{-1}$ gentamycin for 1 h. For measuring the number of bacteria at 1 h, cells were lysed with PBS containing 0.1% Triton X-100 and plated on Luria-Bertani broth plates with appropriate dilutions. For measuring the number of bacteria at 18 h, the DMEM was replaced after 1 h with fresh DMEM containing 12 μg ml$^{-1}$ gentamycin, and the incubation was

continued at 37 °C. After 18 h, cells were lysed with PBS containing 0.1% Triton X-100 and plated on Luria-Bertani broth plates, as indicated above. The percentage survival was obtained by dividing the number of bacteria recovered after 18 h by the number of bacteria present at 1 h. All experiments were performed in duplicate and the results are representative of at least three independent experiments.

**Measuring gene expression inside macrophages.** Gene expression inside macrophages was measured[24] with the following modifications. Macrophage infection was performed as described in the previous section, except that RPMI tissue culture medium was used for RNA extraction. At each time-point, the infected macrophages were lysed and stabilized with Tri reagent (Applied Biosystems) and RNA was extracted according to the manufacturer's instructions. Control RNA was obtained from *Salmonella* grown to exponential growth phase ($OD_{600} = 0.5$) in RPMI tissue culture media. RNA expression in vivo/in vitro was determined by measuring (mRNA levels of each gene inside macrophages/mRNA levels of *rrsH* inside macrophages)/(mRNA levels of each gene in the RPMI media/mRNA levels of *rrsH* grown in the RPMI media).

**Mouse virulence assay.** Six-to-eight-week-old female C3H/HeN mice were inoculated intraperitoneally with ~$10^3$ colony-forming units of *Salmonella* strains. Mouse survival was followed for 21 days. Virulence assays were conducted three times with similar outcomes and the data correspond to groups of five mice. All animals were housed in a temperature- and humidity-controlled room, in which a 12 h light/12 h dark cycle was maintained. All procedures were performed according to the protocols (KW-181010-1) approved by the Institutional Animal Care and Use Committee of the Kangwon National University.

**Measuring non-replicating *Salmonella* inside macrophages.** Macrophage infection[24] and assessment of non-replicating *Salmonella*[70] were performed with the following modifications. J774 A.1 macrophages were grown in Dulbecco modified Eagle medium (DMEM; PAA Laboratories) supplemented with 10% (vol vol$^{-1}$) fetal bovine serum (FBS) and 1% (vol vol$^{-1}$) antibiotic-antimycotic solution at 37 °C and 5% (vol vol$^{-1}$) $CO_2$ conditions in a T75 flask. Prior to infection, $7 \times 10^5$ macrophages were seeded in 24-well plates and incubated with 10% (vol vol$^{-1}$) fetal bovine serum (FBS) and 1% (vol vol$^{-1}$) antibiotic-antimycotic solution at 37 °C and 5% (vol vol$^{-1}$) $CO_2$ conditions for 20 h. *Salmonella* carrying pFCcGi, an mCherry-constitutive and GFP-inducible plasmid, were grown overnight in Luria-Bertani medium with 10 mM L-arabinose to induce GFP expression, and used to infect macrophages at a multiplicity of infection of 10:1. At the indicated time points, infected macrophages were washed and lysed with PBS solution containing 0.1% Triton X-100 (Sigma) to release the intracellular bacteria. The remaining bacterial cells were pelleted and resuspended in PBS solution. The fluorescence of the bacterial population was subsequently assessed by FACS analysis. Samples were analyzed on a NovoCyte$^{TM}$ Flow Cytometer (ACEA) using NovoExpress® software (ACEA). On the NovoCyte$^{TM}$ Flow Cytometer, fluorophores were excited at a wavelength of 488 nm, and green and red fluorescence were detected at 530 and 615 nm, respectively. Data were analyzed with NovoExpress® software. To analyze the fluorescence dilution, bacteria were identified after gating on the constitutive mCherry-positive signal. A grid for chasing the remaining non-replicating cells at the indicated times was drawn based on the GFP$^{High}$ populations of wild-type *Salmonella* at 1 h post infection. The percentage of cells expressing high levels of GFP (GFP$^{High}$) inside J774 A.1 macrophages at the indicated times was calculated based on the following formula: (GFP$^{High}$ mCherry$^{positive}$ *Salmonella* inside the grid)/(total number of *Salmonella* expressing mCherry and GFP fluorescence in Q-2 area, see Supplementary Fig. 8f).

**Reporting summary.** Further information on research design is available in the Nature Research Reporting Summary linked to this article.

## Data availability
RNA sequencing data have been deposited in the Gene Expression Omnibus (GEO) database with accession number GSE103153. All other relevant data are provided as Source Data Files or available from the corresponding author upon reasonable requests.

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

## Acknowledgements

We thank Drs. Yunje Cho and Nam-Chul Ha for their thoughtful comments on PhoR structural modeling. This work was supported by the Basic Science Research Program through the National Research Foundation of Korea (NRF) funded by the Ministry of Science, ICT & Future Planning (NRF-2019R1A2C2003460) and Korea University grants (K1823071 and K1821661) to E.L., and by a grant in the Basic Science Research Program through the National Research Foundation of Korea (NRF) funded by the Ministry of Science, ICT & Future Planning (NRF-2018R1D1A1B07043844) to E.C.

## Author contributions

E.-J.L. designed the research; S.C. and E.C. performed most experiments; Y.-J.C. performed the RNA sequencing and data deposition; D.N. and J.L. performed the initial experiments; Y.-J.C. and E.-J.L. analyzed the data; and E.-J.L. and S.C. wrote the paper.

## Additional information

**Competing interests:** The authors declare no competing interests.

**Peer Review Information:** *Nature Communications* thanks Anne Blanc-Potard and other anonymous reviewer(s) for their contribution to the peer review of this work. Peer reviewer reports are available.

