## [Peer Review File · Nature Communications]

Reviewers' comments:

Reviewer #1 (Remarks to the Author):

The authors observed that high-level ectopic expression of the *mgtC* virulence protein of *Salmonella enterica* serovar Typhimurium results in increased transcription of 72 genes, which most prominently include the *pstSCAB* *phoU* phosphate transport genes and other member of the phosphate-responsive PhoR/PhoB regulon. They show that the MgtC protein interacts with the phosphate-sensing PhoR protein and thereby stimulates its autophosphorylation, resulting in increased transcription of genes in the PhoR/PhoB regulon. They concluded that a specific amino acid in PhoR, Leu421, is required for the interaction with MgtC and for the MgtC-dependent stimulation of phosphate uptake. Furthermore, replacement of Leu421 with Ala resulted in increased survival of *S. Typhimurium* in macrophages and in enhanced virulence in mice.

The basic observations, especially the finding that overproduction of MgtC resulted in increased transcription of genes involved in phosphate homeostasis are important and interesting, and ought to be published. However, I found this paper very difficult to read, and some of the experiments are very difficult to follow because of inadequate description of the experimental methods. I recommend major stylistic revisions for the paper to be accepted.

Specific comments:

1. p. 4, lines 17 to 22: I do not understand why inhibition of the F1F0 ATP synthase would maintain the "cytosolic pH against a large amount of proton (sic) generated during phagosome acidification"?

The F1F0 proton pump is reversible: it can expel protons with energy provided by ATP hydrolysis and it can synthesize ATP powered by the influx of protons. Under acidic external pH, the cells need to expend more ATP's than at neutral pH to pump out protons that leak across the membrane or enter the cells via a number of proton-driven transport systems. So, I don't see why inhibiting the F1F0 ATP synthase /ATPase would help the cells to maintain the intracellular pH near neutrality in face of acidic external pH.

2. p. 6, lines 17 to 19: In view of the report that MgtC promotes the stability of the PhoP protein (Yeom et al. *Molecular Cell* 66: 234-246.e5, 2017), isn't it surprising that the "mRNA levels of the *phoP* and *mgtB* genes were unaffected by *mgtC* overexpression"? The authors should comment in this potential discrepancy.

3. p. 7, second paragraph and Figure 2 legend:

a) explain why *mgtR* is used as positive control

b) Panels C and D: provide better documentation that the bands that are detected were due to the *MgtC* and *PhoR*-GFP proteins. One way this could be done is by showing the entire western blot, including molecular weight markers in Supplementary Information.

Why are two different bands detected with the anti-GFP antibodies, i.e. what is the upper band due to?

4. p. 9, last 2 lines to p. 10, top paragraph, Figs. 3E and 3F, p. 10, 2nd paragraph, and Figs. 4B to 4G: The authors need to explain in the “Methods” section and/or the figure legends how they identified that the radioactive bands were due to phosphorylated-*PhoR*. *Salmonella* has several other membrane-bound sensor kinases (e.g. *PhoQ*, *EnvZ*, *KdpD*, etc). Wouldn't these also be labeled with radioactive phosphate? The autoradiograms of the entire gels with molecular weight standards should be shown in Supplementary Information.

5. p. 11, lines 19 to 21: Why is it that “the *Leu421* to *Gly* substitution had no effect on autophosphorylation of the *PhoR* protein but had a defect in the increase of *phoE* mRNA levels in low phosphate (Fig. S4)”? Could this mutation compromise the ability of *PhoR*-P to transfer its phosphate to *PhoB*?

6. p. 12, first paragraph and Figures 4M and 4N: the phosphate transport data should be expressed as specific activities (moles of phosphate taken up/mg protein/minutes), rather than simply as counts per minute (cpm).

p. 12, lines 11 to 13: If “cells were unable to transport phosphate”, how were they able to grow?

7. p. 14, top paragraph: The statement “this section of data demonstrated that *MgtC*'s regulatory action on expression of phosphate transport genes via *PhoB*/*PhoR* two-component system is required for *Salmonella* pathogenesis” is inconsistent with the statement that “the *phoR* *Leu431* to *Ala* substitution or *phoB* deletion rendered *Salmonella* hypervirulent”.

8. Figures 5A to 5C; explain what “RNA expression *in vivo*/*in vitro*” means and how it was determined.

9. p. 15, lines 12 to 15 and Fig. 5F: What do the authors mean by “GFP-high cells”? Is there a quantitative measure of what “high” is?

The description of how the growth of the Salmonella cells in macrophage was quantified is vague. For example, suppose that the fluorescence was 25% of the starting value after a given time post-infection. This could mean that all of the cells divided 2 times (to reduce the fluorescence per cell 4-fold) or that 75% of the cells grew to the point that their GFP fluorescence was reduced to background, and 25% did not grow at all. Which do the authors mean?

It is not clear why the data in Fig. 5F imply that “MgtC might have multiple targets affecting the formation of non-replicating cells”.

10. p. 20, lines 2 to 5: The text “the mgtC mutant that might have pleiotropic effects on its multiple targets displays a different behavior in fluorescence dilution assay as well as mouse virulence as presented in Figure 5” is unclear and should be reworded.

Reviewer #2 (Remarks to the Author):

The manuscript from Choi et al. describes a novel role for the Salmonella MgtC virulence factor, which is reported to interact with the PhoR histidine kinase and to activate phosphate transport, independently of phosphate concentration. A link is made between this novel function and Salmonella virulence, more specifically non-replicating bacteria that reside inside macrophages.

MgtC is an amazing bacterial factor, present in several pathogens, but mainly studied in Salmonella. It is highly regulated and has a pleiotropic function linked to modulation of F-ATP synthase activity, cellulose synthesis and level of PhoP protein. The additional function described here is of importance and, as suggested by the reviewers, could apply to other pathogens as well, like Mycobacterium tuberculosis.

A large panel of results is shown. Experimental work is well designed and the methodology used in the paper is appropriate but stronger evidence should be required for some conclusions (the causative link between several aspects is not always convincing). Phenotypes associated with PhoRL421A are proposed to be related to the lack of interaction of PhoRL421A with MgtC. Several additional experiments/controls would be required to strengthen this conclusion (see below question about BACTH experiment, lack of experiments in the context of mgtC mutant, puzzling difference between PhoRL421A and PhoRL421G in the ability to respond to low phosphate).

Previous works should be better discussed, including the hypothesis of a decreased phosphate concentration within the bacteria by MgtC (linked to its effect on intracellular ATP) that would activate PhoR (since phosphate sensing via PhoR may occur in the cytoplasm).

Major points

1) The authors provide convincing data to show that the increased of mgtC expression acts to increase expression of PhoBR regulon in a PhoB dependent manner. Several questions arise to better understand the physiological implication of this result (which is issued from artificial overexpression).

- What is the effect increased of mgtC expression on the PhoBR regulon (phoE expression) in conditions inducing PhoR (low phosphate) ? This should be tested with PhoR WT, L421A and L421G

- Conversely, what is the effect of mgtC deletion on the PhoBR regulon (phoE expression and possibly PhoB auto-phosphorylation) in conditions inducing PhoR (low phosphate). This should be tested with PhoR WT, L421A and L421G

In addition, it is not clear why the magnesium concentration is shifted from 10 mM to 0.5 mM ? (mgtC overexpression is driven by IPTG). Is 0.5 mM magnesium considered as PhoQ inducing condition ? (in Fig. 4, low magnesium is 0.01 mM)

2) Authors used bacterial two-hybrid (BACTH) and co-IP to show that MgtC interacts with PhoR and that L421A and L421G substitutions in PhoR prevent this interaction

There are several problems in the use of the BACTH system to convincingly demonstrate in vivo interaction.

PhoR harbors a single TM and a cytoplasmic C-ter (as suggested by Fig 2A and helix prediction websites).

- To monitor in vivo interaction, authors should not use pKT25 which fuses T25 to the N-terminal end of the protein (and it therefore predicted to be in the periplasm). pKNT25 which fuses T25 to C-ter should rather be used. This point should be clarified

- Beta-galactosidase activity should be measured to better appreciate the difference between positive, “strong blue” for PhoR and “weak blue” for PstB (Fig 2B). Similarly, in Fig S3B, authors reported that 421 Ile has no effect whereas it seems different from PhoR WT and 421 Leu
- The interacting part of MgtC should be addressed. Interaction of MgtC with F-ATP synthase has been shown to be dependent on the N92 residue of MgtC, which is located in the TM part of the protein. Does it play a role in interaction with PhoR ? Does a MgtC-T18 recombinant lacking the cytoplasmic C-ter of MgtC still interact with PhoR ?

As mentioned in point 1, it would be informative to test the in vitro phenotypes of L421A mutation independently of mgtC (in a mgtC mutant).

3) The opposite behaviour of PhoR L421A and PhoR L421G in Fig S4G is puzzling: in this figure, PhoR L421A responds perfectly to low phosphate (phoE induction) whereas the response of PhoR L421G is completely abrogated (no induction of phoE). In all other experiments (BACTH, immunoprecipitation, PhoR autophosphorylation, phoE expression in low and high magnesium), both mutants behave similarly. Given the similarity between Alanine and Glycine, authors should carefully confirm the result of Fig S4G and discuss this finding.

4) The phenotype with non-replicating bacteria is difficult to appreciate. Statistical analysis could help, as well as comparison with other mutants known to modulate the number of non-replicating bacteria (at least in the discussion).

What is the rationale for testing at 6h and 9h post infection ? (whereas later time points of 18 to 24 hrs seem to be rather evaluated in the literature).

5) The discussion is sometimes long and important work related to this study is not properly discussed. The results reported by Pontes and Groisman (ref 31, cited in the introduction) should be discussed (it is reported in this study that MgtC activates PhoB by decreasing cytoplasmic Pi levels). The implication of the current findings to other bacteria as *M. tuberculosis* could be discussed taking into account the literature mentioned in the result section that links persisters and phosphate metabolism. Authors should also discuss (dissociate ?) the intramacrophage role of MgtC/F ATP synthase interaction versus MgtC/PhoR interaction. This could be included in the model in Fig 6.

Minor points

- 1) In the abstract, authors mention that MgtC “activates phosphate transport independently of available phosphate concentration during infection”, but this is extrapolated from in vitro findings (similarly in the discussion “activation of phosphate uptake is independent of the concentration of phosphate ions existing in the phagosomal space”)

- 2) In the introduction (p5), authors suggest that Salmonella has to deal with the high mgtC expression within macrophages, which results in a decrease in ATP levels. They propose the present mechanism has a clue. Authors should mention already known mechanisms that contribute to limit the amount of MgtC protein through negative regulation by amgR and MgtR

- 3) In Fig 1A and Fig 4A, PhoR is shown with 2 TM domain. This should be modified if PhoR harbors a single TM, as indicated in the text and predicted by bioinformatics. Moreover, Fig 1A should concentrate on the PhoBR regulon (MgtC and L412 residue should rather belong to the model in Fig 6)

- 4) Loading control would be appreciable in Fig 3 E,F and Fig 4 B, C, D

- 5) For both intramacrophage survival, it would be interesting to test phenotype of a strain that harbors both phoRL421A and mgtC deletion

- 6) Many grammatical errors throughout the manuscript (difficult to list due to the lack of line numbering)

Reviewer #3 (Remarks to the Author):

This study examines the impact of the Salmonella virulence protein MgtC on Salmonella phosphate homeostasis. The authors propose that MgtC activates autophosphorylation of PhoR, which in turn phosphorylates PhoB and activates transcription of PhoB-dependent genes. Surprisingly, this MgtC-dependent activation of the PhoB regulon seems to diminish Salmonella virulence in mice.

Many of the findings of this manuscript have been previously reported by another group (Pontes et al. *Genes Dev.* 2018 Jan 1;32(1):79-92). Although that paper is cited (Reference 31), the similarity of key findings and the striking discrepancies in interpretation are all concealed. In my opinion, this is very poor scholarship.

Most importantly, Pontes et al. already show that overexpression of *mgtC* activates the PhoB-regulon, while *mgtC* mutation weakens PhoB activation. However, they also show that PhoB activation actually precedes MgtC induction. Most importantly, a *mgtC* mutant has actually dramatically higher cytoplasmic phosphate content compared to wild-type *Salmonella*. This increased phosphate concentration is fully sufficient to explain low PhoB activation. If the authors want to argue against this previously published conclusion that seems to be well supported by high-quality data, they need to provide solid evidence that explains the discrepancies.

The finding that a *phoR* L421A mutation also weakens PhoR autophosphorylation as well as interaction with MgtC is interesting and new, but this mutation is potentially difficult to interpret as it also affects at least some MgtC-unrelated responses of PhoR (Fig. 4H, 10 mM Mg; Fig. S4).

The interpretation of Fig. 5F/Fig. S6 is unclear. The lower fraction of persisters for *phoRL421A* and *phoB* could indicate more proliferating bacteria (at least for *phoB* this seems likely based on the histograms in S6), enhanced killing of persisters, etc. Absolute numbers of events in each fraction could help to distinguish between these alternatives.

The mouse virulence data are in conflict with previous findings that a *phoB* mutant is actually attenuated in mice (Becker et al. *Nature.* 2006 Mar 16;440(7082):303-7.), and that a *pstS* mutant (defective for high-affinity phosphate transport) shows wild-type levels of fitness (Valdivia & Falkow *Science.* 1997 Sep 26;277(5334):2007-11.). The authors should provide CFU data from infected organs and assess a complemented mutant.

A native speaker should check the language (e.g., *Salmonella's* phosphate transport is not "designed" to be activated).

Response to Referees

Reviewer #1 (Remarks to the Author):

The authors observed that high-level ectopic expression of the *mgtC* virulence protein of *Salmonella enterica* serovar Typhimurium results in increased transcription of 72 genes, which most prominently include the *pstSCAB phoU* phosphate transport genes and other member of the phosphate-responsive PhoR/PhoB regulon. They show that the MgtC protein interacts with the phosphate-sensing PhoR protein and thereby stimulates its autophosphorylation, resulting in increased transcription of genes in the PhoR/PhoB regulon. They concluded that a specific amino acid in PhoR, Leu421, is required for the interaction with MgtC and for the MgtC-dependent stimulation of phosphate uptake. Furthermore, replacement of Leu421 with Ala resulted in increased survival of *S. Typhimurium* in macrophages and in enhanced virulence in mice.

The basic observations, especially the finding that overproduction of MgtC resulted in increased transcription of genes involved in phosphate homeostasis are important and interesting, and ought to be published. However, I found this paper very difficult to read, and some of the experiments are very difficult to follow because of inadequate description of the experimental methods. I recommend major stylistic revisions for the paper to be accepted.

Specific comments:

1. p. 4, lines 17 to 22: I do not understand why inhibition of the F1F0 ATP synthase would maintain the "cytosolic pH against a large amount of proton (sic) generated during phagosome acidification"?

The F1F0 proton pump is reversible: it can expel protons with energy provided by ATP hydrolysis and it can synthesize ATP powered by the influx of protons. Under acidic external pH, the cells need to expend more ATP's than at neutral pH to pump out protons that leak across the membrane or enter the cells via a number of proton-driven transport systems. So, I don't see why inhibiting the F1F0 ATP synthase /ATPase would help the cells to maintain the intracellular pH near neutrality in face of acidic external pH.

I agree with the reviewer's comment that the F₁F₀ proton pump is reversible. However, we believe that MgtC maintains cytoplasmic pH based on following reasons: Firstly, MgtC's binding to F₁F₀ ATP synthase inhibits proton translocation (Lee et al., Cell 154:146-156, Figure 2). Secondly, MgtC-expressing *Salmonella* decreases intracellular ATP levels and membrane potential that might be required for pumping out protons via a number of proton-driven transport systems (*ibid*, Figures 4 and 5). And finally, *Salmonella* lacking MgtC drops intracellular pH to a greater degree compared to *Salmonella* harboring one when the medium is shifted from neutral pH to acidic pH and therefore the *mgtC* mutant has a lower cytoplasmic pH than MgtC-harboring *Salmonella* inside macrophages (*ibid*, Figure 4).

2. p. 6, lines 17 to 19: In view of the report that MgtC promotes the stability of the PhoP protein (Yeom et al. Molecular Cell 66: 234-246.e5, 2017), isn't it surprising that the "mRNA levels of the *phoP* and *mgtB* genes were unaffected by *mgtC* overexpression"? The authors should comment in this potential discrepancy.

In Yeom et al, the *mgtC* chromosomal mutant exhibited lower levels of the PhoP protein compared to wild-type when bacteria were grown for 5.5 h in N-minimal media containing 15 μM Mg^{2+} (a PhoP-activating condition). In a similar condition (grown for 5 h in N-minimal media containing 10 μM Mg^{2+}) that was used to determine PhoP protein levels, they showed that *phoP* mRNA levels were only two-fold lower in the *mgtC* chromosomal mutant compared to those in wild-type *Salmonella* (Yeom et al. Molecular Cell 66: 234-246, Figure 6). In this manuscript, we measured mRNA levels of the *phoP* and *mgtB* genes when the *mgtC* gene is overexpressed from an IPTG-inducible plasmid for 1 h and, in the condition we tested, *phoP* mRNA levels in cells expressing the *mgtC* gene were similar to cells expressing the empty vector when we analyzed by RNA sequencing and quantitative real-time PCR (Figure S1).

3. p. 7, second paragraph and Figure 2 legend:

a) explain why *mgtR* is used as positive control

As commented by the reviewer, we explained in the text why *mgtR* was used as a positive control.

b) Panels C and D: provide better documentation that the bands that are detected were due to the MgtC and PhoR-GFP proteins. One way this could be done is by showing the entire western blot, including molecular weight markers in Supplementary Information.

Why are two different bands detected with the anti-GFP antibodies, i.e. what is the upper band due to?

Raw images of the western blots with molecular weight markers are provided in the Source Data file. The nature of the upper band in the western blot detected with anti-GFP antibodies is presently unclear. However, the size of PhoR-GFP protein is estimated as 75 kDa, which corresponds to the lower band.

4. p. 9, last 2 lines to p. 10, top paragraph, Figs. 3E and 3F, p. 10, 2nd paragraph, and Figs. 4B to 4G: The authors need to explain in the "Methods" section and/or the figure legends how they identified that the radioactive bands were due to phosphorylated-PhoR. *Salmonella* has several other membrane-bound sensor kinases (e.g. PhoQ, EnvZ, KdpD, etc). Wouldn't these also be labeled with radioactive phosphate? The autoradiograms of the entire gels with molecular weight standards should be shown in Supplementary Information.

The autoradiograms of the entire film are provided in the Source Data file. Our revised manuscript describes how we identified phosphorylated PhoR in the Methods. Membrane vesicles isolated from cells grown for 5 h in N-minimal media containing 0.01 mM Mg^{2+} or cells overexpressing MgtC showed one major radioactive band in the autophosphorylation reaction. And this band corresponds to the radioactive band that increases the intensity in the sample prepared from low Pi media (a PhoB/PhoR-inducing condition) and disappears in the sample prepared from the *phoR* deletion mutant *Salmonella* using the same media. We cut off the lower part of dried gels to remove signals from unincorporated radioactive phosphate before autoradiography.

5. p. 11, lines 19 to 21: Why is it that “the Leu421 to Gly substitution had no effect on autophosphorylation of the PhoR protein but had a defect in the increase of *phoE* mRNA levels in low phosphate (Fig. S4)”? Could this mutation compromise the ability of PhoR-P to transfer its phosphate to PhoB?

The reviewer raises a very interesting question. We agree with the reviewer that the Leu421 to Gly substitution might compromise the ability of PhoR-P to transfer its phosphate to PhoB, which we are currently investigating this possibility and hope to address this question in future studies.

6. p. 12, first paragraph and Figures 4M and 4N: the phosphate transport data should be expressed as specific activities (moles of phosphate taken up/mg protein/minutes), rather than simply as counts per minute (cpm).

p. 12, lines 11 to 13: If “cells were unable to transport phosphate”, how were they able to grow?

As suggested by the reviewer, Figures 4M, 4N, and S7 in the revised manuscript were re-plotted as the y-axis represents specific activities (nmole phosphate/mg protein). We have modified the text to clarify that control cells did not increase phosphate uptake in non-inducing media.

7. p. 14, top paragraph: The statement “this section of data demonstrated that MgtC’s regulatory action on expression of phosphate transport genes via PhoB/PhoR two-component system is required for *Salmonella* pathogenesis” is inconsistent with the statement that “the *phoR* Leu431 to Ala substitution or *phoB* deletion rendered *Salmonella* hypervirulent”.

As suggested by the reviewer, we have modified the text to be consistent with the previous statement.

8. Figures 5A to 5C; explain what “RNA expression in vivo/in vitro” means and how it was determined.

RNA expression in vivo/in vitro represents [mRNA levels inside macrophages/mRNA levels in tissue culture media]. It is determined by measuring (mRNA levels of each gene inside macrophages/mRNA levels of *rrsH* inside macrophages)/(mRNA levels of each gene in RPMI media/mRNA levels of *rrsH* in RPMI media). In our revised manuscript, we stated how we determine gene expression inside macrophages in Methods.

9. p. 15, lines 12 to 15 and Fig. 5F: What do the authors mean by “GFP-high cells”? Is there a quantitative measure of what “high” is?

The description of how the growth of the *Salmonella* cells in macrophage was quantified is vague. For example, suppose that the fluorescence was 25% of the starting value after a given time post-infection. This could mean that all of the cells divided 2 times (to reduce the fluorescence per cell 4-fold) or that 75% of the cells grew to the point that their GFP fluorescence was reduced to background, and 25% did not grow at all. Which do the authors mean?

It is not clear why the data in Fig. 5F imply that “MgtC might have multiple targets affecting the formation of non-replicating cells”.

GFP-high cells represent a portion of bacteria that emit high levels of *gfp* fluorescence

inside macrophages at 6 h or 9 h postinfection. GFP^{high} populations were determined by drawing a grid in wild-type *Salmonella* at the time of invasion (1 h postinfection) to chase the percentage of remaining non-replicating cells inside macrophages at following time points. Because we sorted out single cells and measure *gfp* fluorescence of each bacteria, if GFP^{high} populations are 25% of the cells, it means that 25% of bacteria retains a high intensity of *gfp* fluorescence, suggesting that the detected portion of bacteria is non-replicating. Our revised manuscript now includes a detailed gating strategy for flow cytometric detection in Figure S8.

10. p. 20, lines 2 to 5: The text “the mgtC mutant that might have pleiotropic effects on its multiple targets displays a different behavior in fluorescence dilution assay as well as mouse virulence as presented in Figure 5” is unclear and should be reworded.

The reviewer is right. We removed the text and rewrote the paragraph to improve clarity.

Reviewer #2 (Remarks to the Author):

The manuscript from Choi et al. describes a novel role for the Salmonella MgtC virulence factor, which is reported to interact with the PhoR histidine kinase and to activate phosphate transport, independently of phosphate concentration. A link is made between this novel function and Salmonella virulence, more specifically non-replicating bacteria that reside inside macrophages.

MgtC is an amazing bacterial factor, present in several pathogens, but mainly studied in Salmonella. It is highly regulated and has a pleiotropic function linked to modulation of F-ATP synthase activity, cellulose synthesis and level of PhoP protein. The additional function described here is of importance and, as suggested by the reviewers, could apply to other pathogens as well, like Mycobacterium tuberculosis.

A large panel of results is shown. Experimental work is well designed and the methodology used in the paper is appropriate but stronger evidence should be required for some conclusions (the causative link between several aspects is not always convincing). Phenotypes associated with PhoRL421A are proposed to be related to the lack of interaction of PhoRL421A with MgtC. Several additional experiments/controls would be required to strengthen this conclusion (see below question about BACTH experiment, lack of experiments in the context of mgtC mutant, puzzling difference between PhoRL421A and PhoRL421G in the ability to respond to low phosphate).

Previous works should be better discussed, including the hypothesis of a decreased phosphate concentration within the bacteria by MgtC (linked to its effect on intracellular ATP) that would activate PhoR (since phosphate sensing via PhoR may occur in the cytoplasm).

Major points

1) The authors provide convincing data to show that the increased of mgtC expression acts to increase expression of PhoBR regulon in a PhoB dependent manner. Several questions arise to better understand the physiological implication of this result (which is issued from artificial

overexpression).

- What is the effect increased of *mgtC* expression on the PhoBR regulon (*phoE* expression) in conditions inducing PhoR (low phosphate) ? This should be tested with PhoR WT, L421A and L421G

As suggested by the reviewer, our revised manuscript includes new Figures 4K and 4L showing that *mgtC* expression in a PhoB/PhoR-condition (low phosphate) further increases mRNA levels of the *phoE* gene.

- Conversely, what is the effect of *mgtC* deletion on the PhoBR regulon (*phoE* expression and possibly PhoB auto-phosphorylation) in conditions inducing PhoR (low phosphate). This should be tested with PhoR WT, L421A and L421G

Additionally, new Figures S6G and S6H show that MgtC-dependent *phoE* expression was not observed in the *mgtC* deletion background while low phosphate still induces *phoE* expression in the same background.

In addition, it is not clear why the magnesium concentration is shifted from 10 mM to 0.5 mM ? (*mgtC* overexpression is driven by IPTG). Is 0.5 mM magnesium considered as PhoQ inducing condition? (in Fig. 4, low magnesium is 0.01 mM)

We transferred cells to N-minimal medium media containing 0.5 mM Mg²⁺ to adjust cells to a physiological Mg²⁺ concentration before inducing *mgtC* expression by treating IPTG. mRNA levels of the *phoP* and *mgtB* genes increased after 1 h growth in N-minimal media containing 0.5 mM Mg²⁺ as shown below. However, unlike the PhoB/PhoR-dependent genes, *mgtC* overexpression did not further induce mRNA levels of the *phoP* and *mgtB* genes compared to those harboring the empty vector.

Figure. *mgtC* overexpression has no effect on mRNA levels of the *phoP* and *mgtB* genes (A-B) Relative mRNA levels of the *phoP* (A) and *mgtB* (B) genes in wild-type *Salmonella* harboring the *pmgtC* plasmid or vector. Bacteria were grown for 3 h in N-minimal media containing 10 mM Mg²⁺ and then for an additional 1 h in the same media containing 0.5 mM Mg²⁺ and 0.25 mM IPTG. Samples were taken before transferring to N-minimal media containing 0.5 mM Mg²⁺ (T0) and after an additional growth for 1 h in the same media (T1). Data are represented as mean ± SEM (n=3). Expression levels of target genes were normalized to those of 16S ribosomal RNA *rrsH* gene. Relative mRNA levels represent (target RNA / *rrsH* RNA) × 10000.

2) Authors used bacterial two-hybrid (BACTH) and co-IP to show that MgtC interacts with

PhoR and that L421A and L421G substitutions in PhoR prevent this interaction
There are several problems in the use of the BACTH system to convincingly demonstrate in vivo interaction.

PhoR harbors a single TM and a cytoplasmic C-ter (as suggested by Fig 2A and helix prediction websites).

- To monitor in vivo interaction, authors should not use pKT25 which fuses T25 to the N-terminal end of the protein (and it therefore predicted to be in the periplasm). pKNT25 which fuses T25 to C-ter should rather be used. This point should be clarified

Based on that *Salmonella* PhoR has a 91.18% sequence identity with *Escherichia coli* PhoR (Gardener et al., *BMC Genetics*, 2015, 16, S2) and that pKT25-*phoR* fusion was used to show an interaction between PhoR and PhoU protein in *E. coli* (Gardener et al., *J Bacteriol*, 2014, 196:1741-1752), we believe that *Salmonella* PhoR has two transmembrane domains (10-28, 33-51) and both the N- and C-termini of the PhoR protein are in the cytoplasm.

- Beta-galactosidase activity should be measured to better appreciate the difference between positive, "strong blue" for PhoR and "weak blue" for PstB (Fig 2B). Similarly, in Fig S3B, authors reported that 421 Ile has no effect whereas it seems different from PhoR WT and 421 Leu

As suggested by the reviewer, our revised manuscript includes β -galactosidase activities of bacterial two-hybrid assays in new Figures 2B and S4B.

- The interacting part of MgtC should be addressed. Interaction of MgtC with F-ATP synthase has been shown to be dependent on the N92 residue of MgtC, which is located in the TM part of the protein. Does it play a role in interaction with PhoR? Does a MgtC-T18 recombinant lacking the cytoplasmic C-ter of MgtC still interact with PhoR?

As mentioned in point 1, it would be informative to test the in vitro phenotypes of L421A mutation independently of mgtC (in a mgtC mutant).

Our revised manuscript now includes a new section describing that the N-terminal transmembrane region and Asn92 residue of MgtC are required for PhoR interaction.

3) The opposite behaviour of PhoR L421A and PhoR L421G in Fig S4G is puzzling: in this figure, PhoR L421A responds perfectly to low phosphate (phoE induction) whereas the response of PhoR L421G is completely abrogated (no induction of phoE). In all other experiments (BACTH, immunoprecipitation, PhoR autophosphorylation, phoE expression in low and high magnesium), both mutants behave similarly. Given the similarity between Alanine and Glycine, authors should carefully confirm the result of Fig S4G and discuss this finding.

The opposite behavior of PhoR L421A and PhoR L421G in low phosphate condition is intriguing. As commented by the reviewer 1, the Leu421 to Gly substitution might compromise the phosphotransferase activity of PhoR transferring phosphate from phosphorylated PhoR to PhoB, which we are currently investigating this possibility and hope to address this question in future studies.

4) The phenotype with non-replicating bacteria is difficult to appreciate. Statistical analysis could help, as well as comparison with other mutants known to modulate the number of non-

replicating bacteria (at least in the discussion).

What is the rationale for testing at 6h and 9h post infection ? (whereas later time points of 18 to 24 hrs seem to be rather evaluated in the literature).

As suggested by the reviewer, we carried out a statistical analysis of our data in new Figure 5F. We chose 6 h and 9 h postinfection because mRNA levels of the *mgtC* gene increased at 6 h and 9 h post infection but decreased at 21 h postinfection. To our current knowledge, there is no relevant mutant involved in altering non-replicating *Salmonella* at 6 h or 9 h postinfection and therefore we used the *mgtC* and *phoB* mutant strains in parallel.

5) The discussion is sometimes long and important work related to this study is not properly discussed. The results reported by Pontes and Groisman (ref 31, cited in the introduction) should be discussed (it is reported in this study that MgtC activates PhoB by decreasing cytoplasmic Pi levels). The implication of the current findings to other bacteria as *M. tuberculosis* could be discussed taking into account the literature mentioned in the result section that links persisters and phosphate metabolism. Authors should also discuss (dissociate ?) the intramacrophage role of MgtC/F ATP synthase interaction versus MgtC/PhoR interaction. This could be included in the model in Fig 6.

We thank the reviewer's suggestion. We modified the discussion more concisely and the revised manuscript includes the implication of the current findings in the discussion and in the model of new Figure 6.

As discussed in the revised manuscript, our data unequivocally demonstrate that 1) MgtC activates mRNA levels of the PhoB-dependent genes by directly interacting with PhoR histidine kinase not by indirectly decreasing cytoplasmic Pi levels as proposed by Pontes and Groisman (Genes and Dev 2018, 32:79-92), 2) MgtC exerts its effects on the increase in PhoR autophosphorylation, mRNA levels of the PhoB-dependent genes when expressed from the IPTG-inducible promoter for 1 h, which was tested in high phosphate medium, and 3) MgtC overexpression does increase radioactive phosphate uptake immediately (new Figure S7) but does not alter intracellular phosphate levels during the time we tested (new Figure S9).

Please note that Pontes and Groisman (Genes and Dev 2018, 32:79-92) used low phosphate-MOPS media containing 0.5 mM K_2HPO_4 , whereas we used N-minimal media containing 10 mM KH_2PO_4 in all conditions tested except when we tested a PhoB/PhoR-inducing condition (N-minimal media containing 0.01 mM KH_2PO_4).

Minor points

1) *In the abstract, authors mention that MgtC "activates phosphate transport independently of available phosphate concentration during infection", but this is extrapolated from in vitro findings (similarly in the discussion "activation of phosphate uptake is independent of the concentration of phosphate ions existing in the phagosomal space")*

The reviewer is right. We modified the text in the abstract and discussion accordingly.

2) *In the introduction (p5), authors suggest that Salmonella has to deal with the high mgtC expression within macrophages, which results in a decrease in ATP levels. They propose the present mechanism has a clue. Authors should mention already known mechanisms that contribute to limit the amount of MgtC protein through negative regulation by amgR and MgtR*

We thank the reviewer's suggestion. We modified the introduction accordingly.

3) *In Fig 1A and Fig 4A, PhoR is shown with 2 TM domain. This should be modified if PhoR harbors a single TM, as indicated in the text and predicted by bioinformatics. Moreover, Fig 1A should concentrate on the PhoBR regulon (MgtC and L412 residue should rather belong to the model in Fig 6)*

As discussed above, because we believe that *Salmonella* PhoR has two transmembrane domains (10-28, 33-51) and both the N- and C-termini of the PhoR protein are in the cytoplasm, we prefer to keep as it is.

4) *Loading control would be appreciable in Fig 3 E,F and Fig 4 B, C, D*

As suggested by the reviewer, we provided raw images of the scanned films of Figures 3E and 3F and coomassie stained gels of membrane vesicles prepared for Figure 4B-4D in the Source Data file.

5) *For both intramacrophage survival, it would be interesting to test phenotype of a strain that harbors both phoRL421A and mgtC deletion*

As suggested by the reviewer, our revised manuscript includes new Figure 5D showing the phenotype of the strain with both the *phoR*^{L421A} gene and *mgtC* deletion.

6) *Many grammatical errors throughout the manuscript (difficult to list due to the lack of line numbering)*

We thank the reviewer's suggestion. We polished the language by the editing service to improve readability of the manuscript.

Reviewer #3 (Remarks to the Author):

This study examines the impact of the Salmonella virulence protein MgtC on Salmonella phosphate homeostasis. The authors propose that MgtC activates autophosphorylation of PhoR, which in turn phosphorylates PhoB and activates transcription of PhoB-dependent genes. Surprisingly, this MgtC-dependent activation of the PhoB regulon seems to diminish Salmonella virulence in mice.

Many of the findings of this manuscript have been previously reported by another group (Pontes et al. Genes Dev. 2018 Jan 1;32(1):79-92). Although that paper is cited (Reference 31), the

similarity of key findings and the striking discrepancies in interpretation are all concealed. In my opinion, this is very poor scholarship.

Most importantly, Pontes et al. already show that overexpression of mgtC activates the PhoB-regulon, while mgtC mutation weakens PhoB activation. However, they also show that PhoB activation actually precedes MgtC induction. Most importantly, a mgtC mutant has actually dramatically higher cytoplasmic phosphate content compared to wild-type Salmonella. This increased phosphate concentration is fully sufficient to explain low PhoB activation. If the authors want to argue against this previously published conclusion that seems to be well supported by high-quality data, they need to provide solid evidence that explains the discrepancies.

We thank the reviewer's suggestion. As discussed in the revised manuscript and response to reviewer 2's 5th comment, our data unequivocally demonstrate that 1) MgtC activates mRNA levels of the PhoB-dependent genes by directly interacting with PhoR histidine kinase not by indirectly decreasing cytoplasmic Pi levels as proposed by Pontes and Groisman (Genes and Dev 2018, 32:79-92), 2) MgtC exerts its effects on the increase in PhoR autophosphorylation, mRNA levels of the PhoB-dependent genes when expressed from the IPTG-inducible promoter for 1 h, which was tested in high phosphate medium, and 3) MgtC overexpression does increase radioactive phosphate uptake immediately (new Figure S7) but does not alter intracellular phosphate levels during the time we tested (new Figure S9).

And again, please note that Pontes and Groisman (Genes and Dev 2018, 32:79-92) used low phosphate-MOPS media containing 0.5 mM K_2HPO_4 whereas we used N-minimal media containing 10 mM KH_2PO_4 in all conditions tested except when we tested a PhoB/PhoR-inducing condition (N-minimal media containing 0.01 mM KH_2PO_4).

*The finding that a *phoR* L421A mutation also weakens PhoR autophosphorylation as well as interaction with MgtC is interesting and new, but this mutation is potentially difficult to interpret as it also affects at least some MgtC-unrelated responses of PhoR (Fig. 4H, 10 mM Mg; Fig. S4).*

The nature of the MgtC-unrelated protein bands from samples prepared from high Mg^{2+} media in Figure 4H is presently unclear, which were not detected at mRNA levels prepared from the same samples as shown below.

Figure. The *phoR* Leu421Ala substitution does not alter mRNA levels of the *phoB* gene in high Mg²⁺

Relative mRNA levels of the *phoB* genes in the *phoB*-HA strain with the wild-type *phoR* gene (EN839), the *phoR* derivative with the Leu 421 to Ala substitution (EN966) or the *phoR* derivative with the Leu 421 to Gly substitution (EN1003) grown in N-minimal media containing 0.01 mM or 10 mM Mg²⁺. Bacteria were grown for 5 h in N-minimal media containing 0.01 mM or 10 mM Mg²⁺ as described in Methods. Data are represented as mean ± SEM from three independent experiments. Expression levels of target genes were normalized to that of 16S ribosomal RNA *rrsH* gene.

The interpretation of Fig. 5F/ Fig. S6 is unclear. The lower fraction of persisters for phoRL421A and phoB could indicate more proliferating bacteria (at least for phoB this seems likely based on the histograms in S6), enhanced killing of persisters. etc. Absolute numbers of events in each fraction could help to distinguish between these alternatives.

As suggested by the reviewer, our revised manuscript includes new Figure S8 showing absolute numbers of events in each fraction. However, we are not sure that we could separate two suggested possibilities without further analyses.

The mouse virulence data are in conflict with previous findings that a phoB mutant is actually attenuated in mice (Becker et al. Nature. 2006 Mar 16;440(7082):303-7.), and that a pstS mutant (defective for high-affinity phosphate transport) shows wild-type levels of fitness (Valdivia & Falkow Science. 1997 Sep 26;277(5334):2007-11.). The authors should provide CFU data from infected organs and assess a complemented mutant.

As suggested by the reviewer, we provided below the colony-forming unit (CFU) data from infected spleens and livers of the *phoB* deletion mutant as well as the *phoB*-complemented strain at 6 days postinfection. *phoB* deletion increased a bacterial burden both in the spleen and liver and this phenotype could be complemented by providing the plasmid-encoded *phoB* gene.

Figure. The *Salmonella phoB* mutant is hypervirulent in mice

(A-B) Colony forming units of wild-type (14028s), the *phoB* deletion mutant (KK10), and the *phoB* mutant strains harboring a plasmid with the *phoB* gene (pBAD33-*phoB*) or empty vector (pBAD33) recovered from spleens (A) and livers (B) of C3H/HeN mice at 6 days post-peritoneal infection. C3H/HeN mice were inoculated intraperitoneally with ~2000 colony forming units of the *Salmonella* strains listed above. Bars represent mean values.

A native speaker should check the language (e.g., Salmonella's phosphate transport is not "designed" to be activated).

We thank the reviewer's suggestion. We polished the language by the editing service to improve readability of the manuscript.

We believe to have incorporated the suggestions made by the reviewers. In addition, we provided new Figures 2E and 2F to confirm the MgtC-PhoR interaction in the revised manuscript. Thank you very much for your consideration of our revised manuscript.

Reviewers' comments:

Reviewer #2 (Remarks to the Author):

The revised manuscript from Choi et al. provides additional data to support their conclusions and answered most of the points I have addressed. However, few points are still not clearly discussed and the use of different scales in new/initial figures is puzzling.

As pointed earlier in point 5, and raised also by another reviewer, published work related to the subject was not properly taken into account in the initial version. It is now improved but as indicated in more detail below, there is still a lack of discussion between current results and previous data (especially Pontes & Groisman, 2018 and Lee et al., 2013).

Authors are writing in the abstract “a molecular target(s) of the MgtC protein has not been identified clearly”, suggesting that PhoR would be the first target clearly identified. Interaction with F-ATP synthase (Lee et al., 2013), PhoP and CigR (Yeom et al., 2017 & 2018) have been previously reported based on immunoprecipitation assays. The first work is cited (but the 2 others are not cited). Linked to the known interaction of MgtC with F-ATP synthase (published in Cell, the first author being the last author of the present study), I have asked to test a mutation in MgtC (N92T) that has been specifically implicated in the interaction of MgtC with the AtpB subunit of ATP-synthase (Lee et al., 2013). Intriguingly, in the revised version, authors show that the same mutation also prevent interaction of MgtC with PhoR. This is an important result, which is only shown as supplementary data, and is not mentioned at all in the discussion (this should be discussed p20). Authors should also keep in mind that in bacterial two-hybrid assay, the quantification of beta-galactosidase activity revealed low activity, indicative of very mild interaction between MgtC and PhoR.

In contrast with a previous report (Pontes & Groisman, Genes and Dev, 2018), authors indicate that the intracellular phosphate level is not changed in mgtC mutant. This is an important finding relatively to the present study, which has been added in the revised version, but the result is mentioned only in the discussion (Fig. S9) and is actually not discussed. Authors should provide hypothesis to explain the difference between the two studies.

In addition to these comments, the scale bars of some revised figures is different from the original figures (see PDF Fig4 and Fig5): new Fig4 and S6 have a scale bar that completely differs from the one of initial Fig4. It gives a very different value for same conditions (see for example blue arrow) and may mask some differences. Same for initial and new Fig5 (see PDF).

Minor points: Panels A, B, C in New Fig S6 are important in complement to Fig 4. To have a better understanding of the picture (and behavior of L421G mutation), it would have been interesting to show the pattern in condition of low Mg²⁺ and low phosphate

The novel data regarding effect of phoRLeu421Ala in the mgtC mutant background is not clearly discussed (see p15, the 6 lines in yellow “Because .. PhoR”)

In addition, I've been asked to provide also feedback regarding the answers to reviewer 1's comments:

Authors provide proper answers to most of the points raised by reviewer 1, except for western blotting in Fig 2: upper band recognized by anti-GFP antibodies in Fig 2D still remains unclear and the raw image of the entire gel is not shown (and no molecular weight markers). For the other panel, additional bands are seen on the entire gel but no molecular weight markers are indicated (see joined PDF with Fig2D and corresponding raw image. Reviewers asked for entire western blot including molecular weight markers for Fig2 C & D and authors have written “Raw images of the western blots with molecular weight markers are provided in the source data file”, but this is not the case.

Reviewer #3 (Remarks to the Author):

The revised text addresses some of my questions, but key points remain unclear.

Phosphate uptake Figure S7

This figure shows immediate uptake after addition of addition of ³²P-phosphate, after 1h of mgtC induction. Pontes and Groisman observed similar kinetics for induction of phosphate transporters when they used mgtC overexpression (their Fig. 4B). By contrast, they observed a delayed induction in cells with wild-type levels of mgtC.

It is difficult to imagine how phosphate uptake could be about ten times faster with mgtC overexpression (Fig. S7), without affecting steady state phosphate concentrations (Fig. S9). Where does all this additional phosphate go?

Indeed, Fig. S9 of this manuscript showing unchanged phosphate concentrations, and Fig. 4D of Pontes and Groisman showing dramatically increased phosphate content, are in striking conflict. The authors should measure phosphate content under the same conditions as Pontes and Groisman to clarify if differences between the media cause these dramatically different results.

Persisters Figure S8

The gating seems to be flawed. All Salmonella should express mCherry. Particles with no red fluorescence in Q-3 are background and should not be used in the calculation. The correct persister fraction would thus be "grid" / quadrant Q-2.

The authors now provide absolute numbers but this does not help to distinguish more proliferation vs. lower persister formation/survival. This is because they just recorded 30,000 events (including lots of background in Q-3). Instead, they should record samples containing the same number of infected / lysed macrophages, regardless of the total count of events (no STOP count, but instead measure the entire sample volume; or use beads with a STOP count on bead numbers). This would enable them to calculate the number of persisters per macrophage, and the number of proliferating Salmonella per macrophage.

Response to Referees

Reviewer #2 (Remarks to the Author):

The revised manuscript from Choi et al. provides additional data to support their conclusions and answered most of the points I have addressed. However, few points are still not clearly discussed and the use of different scales in new/initial figures is puzzling.

As pointed earlier in point 5, and raised also by another reviewer, published work related to the subject was not properly taken into account in the initial version. It is now improved but as indicated in more detail below, there is still a lack of discussion between current results and previous data (especially Pontes & Groisman, 2018 and Lee et al., 2013).

Authors are writing in the abstract “a molecular target(s) of the MgtC protein has not been identified clearly”, suggesting that PhoR would be the first target clearly identified. Interaction with F-ATP synthase (Lee et al., 2013), PhoP and CigR (Yeom et al., 2017 & 2018) have been previously reported based on immunoprecipitation assays. The first work is cited (but the 2 others are not cited).

As suggested by the reviewer, we have modified the abstract and cited the previous works (pages 4 and 5).

Linked to the known interaction of MgtC with F-ATP synthase (published in Cell, the first author being the last author of the present study), I have asked to test a mutation in MgtC (N92T) that has been specifically implicated in the interaction of MgtC with the AtpB subunit of ATP-synthase (Lee et al., 2013). Intriguingly, in the revised version, authors show that the same mutation also prevent interaction of MgtC with PhoR. This is an important result, which is only shown as supplementary data, and is not mentioned at all in the discussion (this should be discussed p20). Authors should also keep in mind that in bacterial two-hybrid assay, the quantification of beta-galactosidase activity revealed low activity, indicative of very mild interaction between MgtC and PhoR.

As suggested by the reviewer, we modified the discussion (page 21).

In contrast with a previous report (Pontes & Groisman, Genes and Dev, 2018), authors indicate that the intracellular phosphate level is not changed in mgtC mutant. This is an important finding relatively to the present study, which has been added in the revised version, but the result is mentioned only in the discussion (Fig. S9) and is actually not discussed. Authors should provide hypothesis to explain the difference between the two studies.

We thank the reviewer for the comment. As suggested by the reviewer, we modified the discussion. We initially thought that the discrepancy came from the differences in experimental conditions and the differences in the steady-state levels of phosphate therein. We used N-minimal media containing 10 mM KH_2PO_4 (high phosphate media) in all experiments except when we tested the PhoB/PhoR-inducing condition, whereas Pontes and Groisman (Genes and Dev 2018, 32:79-92) used low phosphate-MOPS media containing 0.5 mM K_2HPO_4 (low phosphate media). However, the reason for the discrepancy is presently unclear because we could not recapitulate the elevated phosphate levels in the *mgtC* mutant in low phosphate media when we repeated our

experiments using experimental conditions as in Pontes & Groisman, 2018. Please see below our responses to reviewer #3.

In addition to these comments, the scale bars of some revised figures is different from the original figures (see PDF Fig4 and Fig5): new Fig4 and S6 have a scale bar that completely differs from the one of initial Fig4. It gives a very different value for same conditions (see for example blue arrow) and may mask some differences. Same for initial and new Fig5 (see PDF).

We thank the reviewer for the comment. The differences in values in the original submission might arise from the experimental differences in the generation of the data presented in Figures 4 and S6. Specifically, the two sets of data were obtained at different times (2016 Dec-2017 Apr vs 2018 Oct-Nov), in different laboratories (Kyung Hee University vs Korea University), and using different pieces of equipment (7300 Real-Time PCR System vs StepOnePlus™ Real-Time PCR system for real-time PCR) due to relocation during the revision process. We modified the method section to acknowledge this comment. In addition, the passage number of cultured J774 A.1 cells might also contribute to the differences in fold replication presented in Figure 5. We have now repeated these experiments, which provide consistent and reproducible data with the results in the present submission.

Minor points: Panels A, B, C in New Fig S6 are important in complement to Fig 4. To have a better understanding of the picture (and behavior of L421G mutation), it would have been interesting to show the pattern in condition of low Mg²⁺ and low phosphate

We thank the reviewer for the comment. We expect that the autophosphorylation behavior of strains grown in low Mg²⁺ and low phosphate would be a summation of those grown in low Mg²⁺ and low phosphate separately based on that mRNA levels of the *phoE* gene increased in low Mg²⁺ and in low phosphate independently and additively (Figure 4K, black). Additionally, the autophosphorylation behavior of the *phoR* L421G in low Mg²⁺ and low phosphate is expected to be similar to that detected in low phosphate because autophosphorylation of the *phoR* L421G substitution mutant is defective in low Mg²⁺ but fully functional in low phosphate (Figures 4 and S6).

*The novel data regarding effect of *phoR*Leu421Ala in the *mgtC* mutant background is not clearly discussed (see p15, the 6 lines in yellow "Because .. PhoR")*

As suggested by the reviewer, we modified the text to improve clarity.

In addition, I've been asked to provide also feedback regarding the answers to reviewer 1's comments:

Authors provide proper answers to most of the points raised by reviewer 1, except for western blotting in Fig 2: upper band recognized by anti-GFP antibodies in Fig 2D still remains unclear and the raw image of the entire gel is not shown (and no molecular weight markers). For the other panel, additional bands are seen on the entire gel but no molecular weight markers are indicated (see joined PDF with Fig2D and corresponding raw image. Reviewers asked for entire western blot including molecular weight markers for Fig2 C & D and authors have written "Raw images of the western blots with molecular weight markers are provided in the source data file", but this is not the case.

We provided a small area of the raw image detected with anti-GFP antibodies in Figure 2D in the previous submission because we cut off the blot above 50 kDa in the molecular weight marker to detect GFP-tagged PhoR proteins. In the revised source data file, we include a larger area of the scanned film with the aligned molecular marker from the gel.

We have provided the rescaled data of Figures 4K (in wild-type background) and S6G (in the *mgtC* mutant background) below. When we rescaled Figure S6G, we could observe low levels of *phoE* mRNA from the strains with the wild-type *phoR* and L421A-substituted *phoR* in the *mgtC* mutant background grown in low Mg^{2+} and high Pi (bottom right). The nature of this mRNA expression is presently unclear because it is *mgtC*-independent and low Mg^{2+} -mediated expression and there is no difference between the strains with the wild-type and L421A-substituted *phoR* genes.

Reviewer #3 (Remarks to the Author):

The revised text addresses some of my questions, but key points remain unclear.

Phosphate uptake Figure S7

This figure shows immediate uptake after addition of addition of ³²P-phosphate, after 1h of *mgtC* induction. Pontes and Groisman observed similar kinetics for induction of phosphate transporters when they used *mgtC* overexpression (their Fig. 4B). By contrast, they observed a delayed induction in cells with wild-type levels of *mgtC*.

Experiments from our Figure S7 and their Figure 4B are clearly different. Firstly, we measured ³²P-phosphate uptake every 20 sec for 100 sec whereas they measured green fluorescence from strains harboring the P_{psl5}-*gfp* plasmid in 30 min intervals after IPTG induction for another ~ 300 min. Secondly, we overexpressed *mgtC* for 1 h in the wild-type cells to measure phosphate uptake but we could not find the corresponding strain (wild-type/ pMgtC) in their Figure 4B for a direct comparison between wild-type pMgtC and wild-type pVector. Thirdly, we grew strains in media containing 10 mM

KH_2PO_4 to measure MgtC-mediated phosphate uptake whereas they grew cells in media containing 0.5 mM K_2HPO_4 . Because MgtC protein production from the chromosomal location takes place after 4 h growth in low Mg^{2+} , $\text{P}_{\text{pstS}}\text{-gfp}$ expression at 2.5 h in the wild-type cells harboring pVector (wild-type pVector) in their Figure 4B possibly resulted from Pi depletion-mediated $\text{P}_{\text{pstS}}\text{-gfp}$ expression.

It is difficult to imagine how phosphate uptake could be about ten times faster with mgtC overexpression (Fig. S7), without affecting steady state phosphate concentrations (Fig. S9).

Where does all this additional phosphate go?

Indeed, Fig. S9 of this manuscript showing unchanged phosphate concentrations, and Fig. 4D of Pontes and Groisman showing dramatically increased phosphate content, are in striking conflict. The authors should measure phosphate content under the same conditions as Pontes and Groisman to clarify if differences between the media cause these dramatically different results.

We thank the reviewer for the comment. Based on our measurements, steady-state levels of intracellular phosphate of cells grown in N-minimal media containing 10 mM KH_2PO_4 are relatively high (258 ~ 940 μM Pi/mg protein) and specific phosphate uptake activities are 6.78 ± 0.86 nmol Pi/mg protein/min when MgtC is expressed from the chromosomal location for 5 h and 6.40 ± 2.34 nmol Pi/mg protein/min when MgtC is expressed from the plasmid for 1 h. Because we expressed MgtC for ~1 h in both conditions, steady-state levels of intracellular phosphate might be less affected by the MgtC-mediated phosphate uptake in these conditions. (Even though it is expressed from the chromosomal location for 5 h in low Mg^{2+} media, MgtC proteins could be detected after 4 h growth (unpublished data and supported by Pontes et al., Mol Cell 64:480-492, Figure 5A) by a regulatory action of the leader RNA on MgtC expression.)

Unfortunately, we could not recapitulate the elevated phosphate levels in the *mgtC* mutant when we tested our strains in media described by Pontes and Groisman, 2018 (MOPS media containing 0.5 mM K_2HPO_4 and 0.01 mM Mg^{2+} , see attached Figures 1B and 2B). As you will see below, we measured intracellular phosphate levels of strains grown in MOPS media containing 0.5 mM K_2HPO_4 using two different detection methods: EnzChek phosphate assay kit (ThermoFisher, E-6646) and the colorimetric detection of phosphomolybdenum blue (used in Pontes and Groisman, 2018, Kanno et al., 2016). As commented by the reviewer, we initially thought that the discrepancy came from the differences in experimental conditions and the differences in the steady-state levels of phosphate in each experimental condition. However, the reason for the discrepancy is presently unclear based on our determinations.

EnzChek phosphate assay kit (ThermoFisher, E-6646)

A

B

C

Figure 1. Intracellular phosphate levels of strains grown in MOPS media containing 0.5 mM K_2HPO_4 by EnzChek phosphate assay kit

(A) Intracellular phosphate levels in either wild-type (14028s) or the *phoB* deletion mutant *Salmonella* (KK10) harboring a plasmid with the *mgtC* gene (*pmgtC*) or the vector. Bacteria were grown for 3 h in MOPS media containing 0.5 mM K_2HPO_4 and 10 mM Mg^{2+} and then for an additional 1 h in the same media containing 0.5 mM K_2HPO_4 , 0.5 mM Mg^{2+} , and 0.25 mM IPTG. Intracellular phosphate levels correspond to micromole of phosphate per mg of total protein (n=6 independent measurements).

(B-C) Intracellular phosphate levels of wild-type *Salmonella* (14028s), the *phoR* chromosomal mutant with Leu421 replaced by Ala codon (EN949) or Gly codon (EN991), the *phoB* deletion mutant (KK10), and the *mgtC* deletion mutant (EL4) grown for 5 h in MOPS media containing 0.5 mM K_2HPO_4 and either 0.01 mM (B) or 10 mM Mg^{2+} (C). Intracellular phosphate levels correspond to micromole of phosphate per mg of total protein (n=6 independent measurements).

Molybdenum blue method (Pontes and Groisman, 2018, Kanno et al., 2016)

Figure 2. Intracellular phosphate levels of strains grown in MOPS media containing 0.5 mM K_2HPO_4 by the molybdenum blue method

(A) Intracellular phosphate levels of in either wild-type (14028s) or the *phoB* deletion mutant *Salmonella* (KK10) harboring a plasmid with the *mgtC* gene (*pmgtC*) or the vector. Bacteria were grown for 3 h in MOPS media containing 0.5 mM K_2HPO_4 and 10 mM Mg^{2+} and then for an additional 1 h in the same media containing 0.5 mM K_2HPO_4 , 0.5 mM Mg^{2+} , and 0.25 mM IPTG. Intracellular phosphate levels correspond to micromole of phosphate per mg of total protein (n=6 independent measurements).

(B-C) Intracellular phosphate levels of wild-type *Salmonella* (14028s), the *phoR* chromosomal mutant with Leu421 replaced by Ala codon (EN949) or Gly codon (EN991), the *phoB* deletion mutant (KK10), and the *mgtC* deletion mutant (EL4) grown for 5 h in MOPS media containing 0.5 mM K_2HPO_4 and either 0.01 mM (B) or 10 mM Mg^{2+} (C). Intracellular phosphate levels correspond to micromole of phosphate per mg of total protein (n=6 independent measurements).

Persisters Figure S8

The gating seems to be flawed. All Salmonella should express mCherry. Particles with no red fluorescence in Q-3 are background and should not be used in the calculation. The correct persister fraction would thus be "grid" / quadrant Q-2.

The authors now provide absolute numbers but this does not help to distinguish more proliferation vs. lower persister formation/survival. This is because they just recorded 30,000 events (including lots of background in Q-3). Instead, they should record samples containing the same number of infected / lysed macrophages, regardless of the total count of events (no STOP count, but instead measure the entire sample volume; or use beads with a STOP count on bead numbers). This would enable them to calculate the number of persisters per macrophage, and the number of proliferating Salmonella per macrophage.

We appreciate the technical comments from the reviewer. We re-plotted Figure 5F as suggested by the reviewer and modified Figure S8, the text, and methods accordingly. As you will see in the revised Figure 5F, even after subtracting $\text{mCherry}^{\text{low}}$ cells as a background, the decreases in the GFP^{high} cells were detected similarly in the *phoR* substitution mutant and the *phoB* deletion mutant.

References

1. Pontes MH, Groisman EA. Protein synthesis controls phosphate homeostasis. *Genes & development* **32**, 79-92 (2018).
2. Pontes MH, Yeom J, Groisman EA. Reducing Ribosome Biosynthesis Promotes Translation during Low Mg(2+) Stress. *Mol Cell* **64**, 480-492 (2016).
3. Kanno S, *et al.* Performance and Limitations of Phosphate Quantification: Guidelines for Plant Biologists. *Plant & cell physiology* **57**, 690-706 (2016).

Reviewers' comments:

Reviewer #2 (Remarks to the Author):

Choi et al have conducted a second round of revision of their manuscript and have further improved their manuscript.

1) Authors were not able to reproduce the strong increase of intracellular phosphate level in *mgtC* mutant that was previously reported by Pontes and Groisman (eventhough a small trend is seen with the molybdenum blue method). The results are shown in the response to reviewer 3 but they should be included as supplementary information in the manuscript to be available to the scientific community. In the current version, they discuss only condition of high phosphate (p 18), but in their hands, steady-state levels of intracellular phosphate are also constant in low phosphate (condition used by Pontes and Groisman). The controversy should appear in the discussion.

2) Regarding a former comment of reviewer 1 on Fig 2C and 2D and further questions that I have addressed, the raw images that are provided still do not convincingly answer the points (see below: molecular weight either unclear or cut & pasted ?). See joined pdf

3) The question about scale of Fig 4K and S6G was because initial figures had a scale up to 1.5 and a value of 1 appeared high comparatively to background. In the first revised version, the scale shifted to 80. Providing now in the response to reviewers a figure with a scale up to 30 does not really address my point, but this was a minor point.

Reviewer #3 (Remarks to the Author):

My questions have been appropriately addressed.

Response to reviewers

Reviewer #2 (Remarks to the Author):

Choi et al have conducted a second round of revision of their manuscript and have further improved their manuscript.

1) Authors were not able to reproduce the strong increase of intracellular phosphate level in mgtC mutant that was previously reported by Pontes and Groisman (eventhough a small trend is seen with the molybdenum blue method). The results are shown in the response to reviewer 3 but they should be included as supplementary information in the manuscript to be available to the scientific community. In the current version, they discuss only condition of high phosphate (p 18), but in their hands, steady-state levels of intracellular phosphate are also constant in low phosphate (condition used by Pontes and Groisman). The controversy should appear in the discussion.

As suggested by the reviewer, we included the result showing that steady-state levels of intracellular phosphate are relatively constant even in low phosphate medium in new Figure S9 and modified the text accordingly.

2) Regarding a former comment of reviewer 1 on Fig 2C and 2D and further questions that I have addressed, the raw images that are provided still do not convincingly answer the points (see below: molecular weight either unclear or cut & pasted ?). See joined pdf

As suggested by the reviewer, we included the scanned original film (for anti-GFP antibodies) and two images captured by Alliance LD4 chemiluminescence detection system (for anti-MgtC and anti-His antibodies) in the revised Source Data file. Again, we cut the blot into three pieces to detect the immunoprecipitated samples (upper to detect GFP-tagged PhoR proteins, lower right to detect MgtC proteins, and lower left to detect His-tagged PhoU proteins) and there are no cross-reactive bands in the molecular weight markers with those antibodies. Therefore, we provided the blot image (before cutting) to align with the molecular markers. Band signals from anti-His antibodies were weak thus we provided the saturated image to mark the edge of the blot together with the original image. Although we provided the blot image, please understand that the interaction between PhoR and PhoU proteins is beyond this manuscript.

3) The question about scale of Fig 4K and S6G was because initial figures had a scale up to 1.5 and a value of 1 appeared high comparatively to background. In the first revised version, the scale shifted to 80. Providing now in the response to reviewers a figure with a scale up to 30 does not really addressed my point, but this was a minor point.

We agree with the reviewer. In Figure S6G, the mRNA levels of *phoE* gene from the strains with the wild-type and L421A-substituted *phoR* gene in the *mgtC* mutant background grown in low Mg^{2+} and high Pi are relatively high (values of 0.83 and 0.89) compared to those grown in high Mg^{2+} and high Pi (values between 0.04 and 0.06). And, the expression patterns of the graph in Figure S6G looked similar even when we rescaled further. However, again the nature of this mRNA expression is presently unclear because it is *mgtC*-independent and low Mg^{2+} -mediated *phoE* expression and there is no difference between the strains with the wild-type and L421A-substituted *phoR* genes.

Reviewer #3 (Remarks to the Author):

My questions have been appropriately addressed.

We appreciate all the comments made by the reviewer.

REVIEWERS' COMMENTS:

Reviewer #2 (Remarks to the Author):

My concerns have been appropriately addressed